# FFTArray: A Python Library for the Implementation of Discretized Multi-Dimensional Fourier Transforms

Stefan J. Seckmeyer[1][*], Christian Struckmann[1], Gabriel Müller[1], Jan-Niclas Kirsten-Siemß[1] and Naceur Gaaloul[1][†]

**1** Institut für Quantenoptik, Leibniz Universität Hannover, Welfengarten 1, D-30167, Hannover, Germany

[*] seckmeyer@iqo.uni-hannover.de , [†] gaaloul@iqo.uni-hannover.de

## Abstract

Partial differential equations describing the dynamics of physical systems rarely have closed-form solutions. Fourier spectral methods, which use Fast Fourier Transforms (FFTs) to approximate solutions, are a common approach to solving these equations. However, mapping Fourier integrals to discrete FFTs is not straightforward, as the selection of the grid as well as the coordinate-dependent phase and scaling factors require special care. Moreover, most software packages that deal with this step integrate it tightly into their full-stack implementations. Such an integrated design sacrifices generality, making it difficult to adapt to new coordinate systems, boundary conditions, or problem-specific requirements. To address these challenges, we present FFTArray, a Python library that automates the general discretization of Fourier transforms. Its purpose is to reduce the barriers to developing high-performance, maintainable code for pseudo-spectral Fourier methods. Its interface enables the direct translation of textbook equations and complex research problems into code, and its modular design scales naturally to multiple dimensions. This makes the definition of valid coordinate grids straightforward, while coordinate grid specific corrections are applied with minimal impact on computational performance. Built on the Python Array API Standard, FFTArray integrates seamlessly with array backends like NumPy, JAX and PyTorch and supports Graphics Processing Unit acceleration. The code is openly available at https://github.com/QSTheory/fftarray under Apache-2.0 license.

# 1  Introduction

Many interesting and important physical systems are modeled by differential equations that often lack closed-form solutions. Fourier integrals often allow to formulate approximate solutions for these systems. Although performing these integrals analytically may still remain

## Traditional ⇒ With FFTArray

Figure 1: Conceptual overview of FFTArray's role in scientific software architecture. Traditional software couples the definition of the physics problem, the Fourier-based differential equation solver and the discretization of Fourier transforms with Fast Fourier Transforms (FFTs) within a monolithic framework. These components interact with each other via problem-specific internal APIs, creating complex interdependent codebases. FFTArray decouples the implementation of discretized Fourier transforms from the solver and physics problem. This architectural simplification enables researchers to focus on core physics and solver logic without managing low-level FFT implementation details, resulting in more maintainable and reusable scientific code.

intractable, they can often be evaluated numerically by discretizing them and using Fast Fourier Transform (FFT) algorithms.

A prominent example is the Schrödinger equation, which governs the evolution of quantum mechanical systems ranging from single-particle dynamics to many-body phenomena. While exact solutions exist for idealized cases like box-shaped potentials and the quantum harmonic oscillator, most practical applications require numerical methods such as the split-step Fourier method [1, 2]. These numerical simulations allow the validation and improvement of analytical solutions as well as to go beyond our current understanding of complex physical systems.

Implementing a spectral Fourier solver for a particular problem can be split into three parts (fig. 1). At the input level, the user defines their system, representative of a concrete set of potentially time-dependent differential equations. In the case of the Schrödinger equation, this could be the initial state and the Hamiltonian of the system. The solver algorithm transforms these into a list of discrete steps to compute the current state of the system from its previous state. In the case of a spectral Fourier method, like a split-step Fourier solver, each of these steps contains multiple analytical Fourier integrals. These analytical expressions are discretized over a finite spatial volume with a finite sample spacing in order to evaluate them numerically. This involves the translation of the analytical Fourier integrals into Fast Fourier Transforms and must take into account the concrete system, the requirements of the solver as well as the coupling between the sample spacing in position and frequency space, described by the Sampling Theorem [3]. These three blocks are built on top of a general numerical array library, which in the case of Python are libraries like NumPy, JAX and PyTorch [4–6].

Existing implementations often entangle these logically distinct blocks - physical system definition, solver algorithms, and FFT-based discretization (fig. 1) - into application-specific frameworks [7–11]. This monolithic design creates three major limitations:

1. **Lack of generality**: Each implementation needs to focus on a specific subset of systems in order to keep complexity in check. Examples for such limitations are only simulating a single wave function or supporting only exactly two dimensions.

2. **Code duplication**: Due to the limited scope, one needs a new implementation for each significant change in the simulated system or used solver. Since each implementation might use special properties like symmetric grids to simplify their use of FFTs, the discretization of the analytic Fourier transforms into FFTs is re-derived and implemented.

3. **Code complexity**: Tight coupling between implementation shortcuts and special-case logic makes the code diverge significantly from its original mathematical formulation. Any modification to one part of the system risks breaking interconnected components, requiring coordinated updates and understanding across the entire codebase.

We have generally observed that managing these limitations presents a substantial challenge in the development of scientific simulations as systems grow in complexity. To conclude, integrating system definitions, solvers, and Fourier transforms into a monolithic program with narrow scope makes these programs challenging to maintain, reuse or adapt to new problems.

Here, we present the Python library FFTArray, which implements the numerical approximation of a Fourier transform with an FFT on arbitrarily placed multi-dimensional grids. Encapsulating this level cleanly allows numerical simulations to focus on the other two blocks and therefore be significantly simpler (as illustrated in fig. 1). The design and implementation of FFTArray focuses on three goals:

1. **From formulas to code**: Physicists can directly map analytical equations involving Fourier transforms to code without mixing discretization details with physics. This enables rapid prototyping of diverse physical models and solver strategies.

2. **State-of-the-art performance**: We achieve state-of-the-art performance on Graphics Processing Units (GPUs) when solving the Schrödinger equation via split-step algorithms for large quantum systems with more than $10^9$ samples and more than $10^4$ time steps.

3. **Seamless multidimensionality**: Dimensions are broadcast by name which enables a uniform API to seamlessly transition from single- to multi-dimensional systems.

FFTArray is a pure Python library built on top of the Python Array API standard in order to be accessible and maintainable in the future. This design ensures compatibility with the ubiquitous NumPy for smaller calculations in any environment, while also enabling GPU acceleration with JAX and PyTorch.

Despite its broad applicability, we note that FFTArray was developed in the context of simulating the atom-light interaction, dynamics and interferometry of ultracold atomic ensembles, particularly Bose-Einstein condensates (BECs) [12]. Thus, our examples and benchmarking focus on problems within this domain. The time evolution of these systems is determined by a nonlinear Schrödinger equation, referred to as the Gross–Pitaevskii equation (GPE) [13–15]. A concrete system can be simulated by solving this equation using the split-step method [16].

This paper is organized as follows: Section 2 introduces the mathematical framework for approximating the continuous Fourier transform on arbitrary coordinate grids using FFTs. It also highlights special cases of coordinate grid choices which are often chosen to simplify the use of the FFT. Section 3 outlines the rationale behind the design of the FFTArray library that allows near native performance and facilitate seamless interaction with various array libraries. In section 4, we present multiple application examples, ranging from performing a derivative to computing the quantum mechanical ground state of a coupled two-species mixture of Bose-Einstein condensates confined in a harmonic potential. We analyze the computational precision of FFTArray by comparing it with the analytical solution of the single-species isotropic quantum harmonic oscillator. Section 5 showcases the performance of FFTArray using NumPy and JAX on various central processing units (CPUs) and GPUs. This chapter also serves as a starting guide for developing efficient implementations of different solver algorithms based on FFTArray.

## 2 Discretization of the Fourier Transform

The Fourier transform converts an analytical function from its original domain $x$ into its representation in the domain of the conjugate variable $f$, which we will refer to as the frequency domain. The unit of $f$ is given by the inverse of the unit of the variable $x$ in the original domain $[f] = [x]^{-1}$. For a function $g(x) : \mathbb{R} \to \mathbb{C}$ (including $\mathbb{R} \to \mathbb{R}$), defined in the original domain, the Fourier transform $\mathcal{F}$ returns a complex-valued function $G(f) : \mathbb{R} \to \mathbb{C}$ in the frequency domain. This function gives the amplitudes and phases for all frequencies $f$ which make up the original function. The inverse Fourier transform $\widehat{\mathcal{F}}$ converts a function from the frequency domain back to the original domain:

$$\mathcal{F}: \; G(f) = \int_{-\infty}^{\infty} dx \; g(x) \, e^{-2\pi i f x}, \quad \forall \, f \in \mathbb{R}, \tag{1}$$

$$\widehat{\mathcal{F}}: \; g(x) = \int_{-\infty}^{\infty} df \; G(f) \, e^{2\pi i f x}, \quad \forall \, x \in \mathbb{R}. \tag{2}$$

Note, that this definition of the Fourier transform uses the linear frequency $f$ as opposed to the circular frequency $\omega = 2\pi f$. The FFTArray library chooses the name "position space" for the original domain, which we will use from here on. This does not affect its applicability to Fourier transforms in time, where the position space is typically called "time domain".

For many functions it is not feasible to evaluate their Fourier transform analytically. Beyond trivial or well-known analytical functions it is often impractical to calculate the integral in eq. (1) analytically. Moreover, the sampled values of any measurement of a physical quantity or the output of a numerical algorithm do not have an analytical expression to perform the Fourier transform on.

### 2.1 A Discretized Fourier Transform

For the cases where analytically evaluating the Fourier transform is not feasible, one can construct a discretized analog of the Fourier transform. In order to do that, the function is sampled on a finite grid with $N$ equidistant samples $x_n$ in position space and $f_m$ in the frequency space:

$$x_n := x_{\min} + n\Delta x, \quad n = 0, \ldots, N-1, \tag{3}$$

$$f_m := f_{\min} + m\Delta f, \quad m = 0, \ldots, N-1, \tag{4}$$

$$g_n := g(x_n), \tag{5}$$

$$G_m := G(f_m), \tag{6}$$

where the sample spacings $\Delta x, \Delta f > 0$ describe the distance between two samples and $x_{\min}, f_{\min}$ are the smallest samples in position and frequency space, respectively. Using these definitions, the integrals from eq. (1) and eq. (2) can be approximated as finite sums leading to a general discretized Fourier transform (gdFT and gdIFT):

$$G_m = \Delta x \sum_{n=0}^{N-1} g_n \, e^{-2\pi i \, f_m x_n} \quad \text{(gdFT)}, \tag{7}$$

$$g_n = \Delta f \sum_{m=0}^{N-1} G_m \, e^{2\pi i \, f_m x_n} \quad \text{(gdIFT)}. \tag{8}$$

We emphasize that discretizing a continuous function and the Fourier transform has several non-trivial implications. While we only highlight the most important aspects in the context of this work, we refer the reader to ref. [17] for an extended discussion. Technically,

the discretization describes the projection of the original function pair $g(x_n) : \mathbb{R} \to \mathbb{C}$ and $G(f_m) : \mathbb{R} \to \mathbb{C}$ into an expansion of the continuous function as a finite linear combination of trigonometric functions with complex Fourier coefficients. If the function can be described exactly as such a linear combination, this projection is lossless. Otherwise the discretized approximation will be subject to aliasing [3, 17]. A discretization in position space results in a finite amount of frequencies that can be distinguished. Any frequency component that is not covered by one of the sampled frequencies will appear as an additional amplitude and phase in one of the frequencies $f_m$. This phenomenon of aliasing is described in the Nyquist Shannon Sampling Theorem [3]. Analogously, if the discretization is performed in frequency space these aliases appear in position space. Therefore, in the case of a lossy projection it makes a difference in which space the discretization was performed. The correct handling of aliasing heavily depends on the goal. In physics simulations it can often be utilized as periodic boundary conditions or the functions are gratuitously extended at the domain boundaries with zero values to approximate an open domain.

From the sampling theorem follows a relationship between the sampling rate of the position (frequency) space and the periodicity of frequency (position) space which needs to be fulfilled[1]:

$$x_{\text{period}} := N\Delta x = \frac{1}{\Delta f}, \tag{9}$$

$$f_{\text{period}} := N\Delta f = \frac{1}{\Delta x}, \tag{10}$$

$$1 = N\Delta f \Delta x. \tag{11}$$

The maximum range of frequencies that can be distinguished with a sample spacing of $\Delta x$ is also exactly $1/\Delta x$.

Note, that these observations do not constrain the choice of the offsets $x_{\text{min}}$ and $f_{\text{min}}$. When approximating a non-periodic analytic function they need to be chosen such that they minimize aliasing by making sure that the original function is as close to zero as possible outside the intervals $[x_{\text{min}}, x_{\text{min}} + x_{\text{period}}]$ and $[f_{\text{min}}, f_{\text{min}} + f_{\text{period}}]$. In the case of a real valued function g(x), the frequency extent turns into an upper band limit which is called the Nyquist frequency. The Fourier transform $G(f) : \mathbb{R} \to \mathbb{C}$ of a real-valued function $g(x) : \mathbb{R} \to \mathbb{R}$ is conjugate symmetric, i.e., $G(f) = \overline{G(-f)}$. Therefore without additional knowledge about a lower band limit, the frequency window must be placed symmetrically around zero to miminize aliasing. This turns the extent of the frequency window $f_{\text{period}}$ for correct reconstruction into an upper band limit of $f_{\text{Nyquist}} := f_{\text{period}}/2 = 1/(2\Delta x)$. There are other cases where an asymmetric window can be beneficial since it is known that only specific frequencies appear in the sampled function. One example is a quantum mechanical wavefunction describing a Gaussian Wave Packet with a non-zero velocity.

## 2.2 Implementation

The general discretized Fourier transform, eqs. (7) and (8), could be implemented directly and then used as a discrete analog to the continuous Fourier transform. But in practice, there are multiple optimizations possible to ensure the best computational performance. The most important optimization is to replace the naive sums by using Fast Fourier Transform (FFT) algorithms. The resulting expression can then often be simplified even further which is described in section 2.3.

From this point on, we extend the mathematical notation by functions which take an array as the input and return an array. The appearance of an index like $n$ inside the argument of

---

[1]Equation (11) is for $f$ being in the unit of cycles. Without loss of generality it can be substituted by a variable in any unit. For example for an angular frequency $k = 2\pi f$ eq. (11) would be $2\pi = N\Delta k\Delta x$

the function $\mathtt{dft}_m(g_n)$ marks $\mathtt{dft}$ as a function which takes an array in position space as an argument and returns an array in frequency space. The subscript $n$ or $m$ denotes the space of their result.

$$\mathtt{dft}_m(g_n) \equiv \mathtt{dft}_m\big(\{g_n\}_{n=0}^{N-1}\big) := \sum_{n=0}^{N-1} g_n \, e^{-2\pi i \, \frac{mn}{N}}, \tag{12}$$

$$\mathtt{idft}_n(G_m) \equiv \mathtt{idft}_n\big(\{G_m\}_{n=0}^{N-1}\big) := \frac{1}{N}\sum_{m=0}^{N-1} G_m \, e^{+2\pi i \, \frac{mn}{N}}. \tag{13}$$

The DFT and inverse DFT have multiple conventions for phase and scaling factors. The above definitions follow NumPy [4], the standard library of scientific computing in Python. Evaluating these sums naively has a computational time complexity of $\mathcal{O}(N^2)$. They can be computed more efficiently with Fast Fourier transform (FFT) algorithms in $\mathcal{O}(N \log N)$ time.

To be able to calculate eqs. (7) and (8) in $\mathcal{O}(N \log N)$ steps we decompose them into separate phase factors and express them in terms of the $\mathtt{fft}$ and $\mathtt{ifft}$:

$$(\text{gdFT}) \quad G_m = \Delta x \sum_{n=0}^{N-1} g_n \, e^{-2\pi i \, (f_{\min}+m\Delta f)(x_{\min}+n\Delta x)} \tag{14}$$

$$= \Delta x \, e^{-2\pi i \, x_{\min} \, m\Delta f} \, e^{-2\pi i \, x_{\min} \, f_{\min}} \sum_{n=0}^{N-1} g_n \, \underbrace{e^{-2\pi i \, m\Delta f \, n\Delta x}}_{e^{-2\pi i \, \frac{mn}{N}}} e^{-2\pi i \, f_{\min} \, n\Delta x} \tag{15}$$

$$= \Delta x \, e^{-2\pi i \, x_{\min} \, m\Delta f} \, e^{-2\pi i \, x_{\min} \, f_{\min}} \, \mathtt{fft}_m\big(g_n \, e^{-2\pi i \, f_{\min} \, n\Delta x}\big), \tag{16}$$

$$(\text{gdIFT}) \quad g_n = \Delta f \sum_{m=0}^{N-1} G_m \, e^{2\pi i \, (f_{\min}+m\Delta f)(x_{\min}+n\Delta x)} \tag{17}$$

$$= \Delta f \, e^{+2\pi i \, f_{\min} \, n\Delta x} \sum_{m=0}^{N-1} G_m \, \underbrace{e^{+2\pi i \, m\Delta f \, n\Delta x}}_{e^{+2\pi i \, \frac{mn}{N}}} e^{+2\pi i \, x_{\min} \, m\Delta f} \, e^{+2\pi i \, x_{\min} \, f_{\min}} \tag{18}$$

$$= e^{+2\pi i \, f_{\min} \, n\Delta x} \, \Delta f \, N \, \mathtt{ifft}_m\big(G_m \, e^{+2\pi i \, x_{\min} \, m\Delta f} \, e^{+2\pi i \, x_{\min} \, f_{\min}}\big)$$

$$= e^{+2\pi i \, f_{\min} \, n\Delta x} \, \mathtt{ifft}_m\big(G_m \, e^{+2\pi i \, x_{\min} \, m\Delta f} \, e^{+2\pi i \, x_{\min} \, f_{\min}}/\Delta x\big). \tag{19}$$

Identical colors highlight exponentials with opposite signs for the forward and backward transforms. The grouping of the factors and the use of eq. (11) in the last step makes the forward and backward transform symmetric which becomes important in section 3.3. All remaining phase factors only depend on at most one of the indices and can therefore be applied to the values before or after the transform in linear time $\mathcal{O}(N)$. Therefore the complete algorithm has a runtime complexity of $\mathcal{O}(N \log N)$.

## 2.3 Special Cases

In many common applications it is possible to remove some of the additional phase terms while still getting correct results for that use case. In this section we discuss some of these special cases and the simplifications they allow. This then also shows the use-cases for the common helper functions for FFTs $\mathtt{fftshift}$, $\mathtt{ifftshift}$ and $\mathtt{fftfreq}$ as defined in NumPy [18]. These functions implement special cases of the general phase factors which are used by FFT-Array. Therefore when using the gd(I)FT via FFTArray these functions are not needed. To properly make the connection between FFTArray and many existing tutorials and implementations, we show in this section how the gd(I)FT can be rewritten in terms of these functions in some special cases of coordinate grids.

The factors $\exp(\pm 2\pi i\ x_{\min}\ m\Delta f)$ and $\exp(\pm 2\pi i\ f_{\min}\ n\Delta x)$ are special cases of shifting a function. Multiplying a function with the factor $\exp(+2\pi i\ f_{\min} x)$ in position space or $\exp(-2\pi i\ x_{\min} f)$ in frequency space shifts that function in the other space by $f_{\min}$ or $x_{\min}$ respectively. Since eqs. (16) and (19) operate on periodic functions, these shifts are cyclic. Any part of the function that is shifted out of the position or frequency window on one side directly reappears on the other side. If $x_{\min}$ or $f_{\min}$ are integer multiples of $\Delta x$ or $\Delta f$, their shifts can be replaced by cyclically shifting the values in the array. Any values which move beyond the end of the array are moved back to the beginning. The functions `fftshift`, `ifftshift` implement such cyclic shifts for half the length of the domain.

### 2.3.1 Symmetric Frequency Space and $x_{\min} = 0$

Sampling a real-valued function is a common special case that can be found in many tutorials. Starting the sampling at $x_{\min} = 0$ is often a natural choice like for example in a time series. With $x_{\min} = 0$, the gd(I)FT simplifies to:

$$\text{(gdFT with } x_{\min} = 0)\quad G_m = \Delta x\ \texttt{fft}_m\left(g_n\ e^{-2\pi i\ f_{\min}\ n\Delta x}\right),$$
$$\text{(gdIFT with } x_{\min} = 0)\quad g_n = e^{+2\pi i\ f_{\min}\ n\Delta x}\ \texttt{ifft}_m\left(G_m/\Delta x\right).$$

As mentioned earlier the frequency space representation of such a function is conjugate symmetric and therefore frequency space must be chosen symmetrically:

$$f_{\min}^{\text{sym}} = -\text{floor}(0.5N)\,\Delta f. \tag{20}$$

Since $f_{\min}^{\text{sym}}$ is an integer multiple of $\Delta f$, the exponential $e^{\pm 2\pi i\ f_{\min}\ n\Delta x}$ can be replaced by a simple shift of the values. These shifts by $f_{\min}^{\text{sym}}$ are implemented in `fftshift` and its inverse `ifftshift`. Replacing the remaining phase factors with these functions reduces the gd(I)FT to the more commonly known form:

$$\text{(gdFT with } x_{\min} = 0, f_{\min} = f_{\min}^{\text{sym}})\quad G_m = \Delta x\ \texttt{fftshift}_m(\texttt{fft}_m(g_n)),$$
$$\text{(gdIFT with } x_{\min} = 0, f_{\min} = f_{\min}^{\text{sym}})\quad g_n = \texttt{ifft}_m(\texttt{ifftshift}_m(G_m/\Delta x)).$$

### 2.3.2 Symmetric Position and Frequency Space

The special case of a position and frequency space, both symmetric around zero with zero being explicitly sampled allows to replace all phase factors with `fftshift` and `ifftshift` as well. A subtlety of this case is that $x = 0$ and $f = 0$ need to be sampled explicitly regardless of whether $N$ is even or odd, analogous to the symmetric frequency space case:

$$x_{\min}^{\text{sym}} = -\text{floor}(0.5N)\,\Delta x, \tag{21}$$
$$f_{\min}^{\text{sym}} = -\text{floor}(0.5N)\,\Delta f. \tag{22}$$

Note that the samples in both position and frequency space with these choices for $x_{\min}$ and $f_{\min}$ are not actually symmetric for even $N$. Recalling eq. (3) and eq. (4) shows that there is one more negative than positive coordinate value. With the coordinates properly chosen as in eqs. (21) and (22) the gd(I)FT can be written as:

$$\text{(gdFT with } x_{\min} = x_{\min}^{\text{sym}}, f_{\min} = f_{\min}^{\text{sym}})\quad G_m = \Delta x\ \texttt{fftshift}_m(\texttt{fft}_m(\texttt{ifftshift}_n(g_n))),$$
$$\text{(gdIFT with } x_{\min} = x_{\min}^{\text{sym}}, f_{\min} = f_{\min}^{\text{sym}})\quad g_n = \texttt{fftshift}_n(\texttt{ifft}_m(\texttt{ifftshift}_m(G_m/\Delta x))).$$

### 2.3.3 Convolution

Another common case where parts of the gd(I)FT can be simplified is the convolution. The convolution has many different applications ranging from image processing over statistics to physics [17, 19, 20]. It can be expressed with (inverse) Fourier transforms via the convolution theorem:

$$g(x) * h(x) = \int_{-\infty}^{\infty} g(\tau) h(x - \tau) \, d\tau \tag{23}$$

$$= \widehat{\mathcal{F}} \{ \mathcal{F}[g(x)] \, \mathcal{F}[h(x)] \} . \tag{24}$$

This enables its computation on discretized data in $\mathcal{O}(N \log N)$ via the FFT. In the general case of arbitrary $x_{\min}$ and $f_{\min}$, one set of the phase and scale factors in frequency space cancels out. However, another set remains because there are two gdFTs and only one gdIFT:

$$
\begin{aligned}
g_n * h_n &= \text{gdIFT}_n \left( \text{gdFT}_m(g_n) \, \text{gdFT}_m(h_n) \right) \\
&= e^{+2\pi i \, f_{\min} \, n \Delta x} \, \text{ifft}_m [ \\
&\quad \Delta x \, e^{-2\pi i \, x_{\min} \, m \Delta f} \, e^{-2\pi i \, x_{\min} \, f_{\min}} \, \text{fft}_m \left( h_n \, e^{-2\pi i \, f_{\min} \, n \Delta x} \right) \\
&\quad \Delta x \, e^{-2\pi i \, x_{\min} \, m \Delta f} \, e^{-2\pi i \, x_{\min} \, f_{\min}} \, \text{fft}_m \left( g_n \, e^{-2\pi i \, f_{\min} \, n \Delta x} \right) \\
&\quad e^{+2\pi i \, x_{\min} \, m \Delta f} \, e^{+2\pi i \, x_{\min} \, f_{\min}} \frac{1}{\Delta x} \\
&\quad ] \\
&= e^{+2\pi i \, f_{\min} \, n \Delta x} \, \text{ifft}_m ( \\
&\quad \Delta x \, e^{-2\pi i \, x_{\min} \, m \Delta f} \, e^{-2\pi i \, x_{\min} \, f_{\min}} \, \text{fft}_m \left( g_n \, e^{-2\pi i \, f_{\min} \, n \Delta x} \right) \text{fft}_m \left( h_n \, e^{-2\pi i \, f_{\min} \, n \Delta x} \right) \\
&\quad ) .
\end{aligned}
$$

In this case it is important to actually apply these remaining phase and scale factors correctly to get the expected result. For special cases of $x_{\min}$ and $f_{\min}$ these shifts can again be replaced by `fftshift`, `ifftshift` or the identity as shown in the other examples.

### 2.3.4 Derivative

Computing a derivative via the Fourier transform can be viewed as a special case of the convolution. In this case one of the convolved functions can be constructed directly in frequency space (for details cf. section 4.1):

$$\frac{d}{dx} g(x) = \widehat{\mathcal{F}} \{ (2\pi i f) \, \mathcal{F} \{ g(x) \} \} . \tag{25}$$

Discretizing this using the gd(I)FT with $x_{\min} = 0$ removes most phase factors:

$$\frac{d}{dx} g_n = \text{gdIFT}_n \left( 2\pi i \, (f_{\min} + m \Delta f) \, \text{gdFT}_m(g_n) \right) \tag{26}$$

$$= e^{+2\pi i \, f_{\min} \, n \Delta x} \, \text{ifft}_m \left( 2\pi i \, (f_{\min} + m \Delta f) \, \text{fft}_m \left( g_n \, e^{-2\pi i \, f_{\min} \, n \Delta x} \right) \right) . \tag{27}$$

Choosing a frequency space symmetric around zero with $f_{\min} = f_{\min}^{\text{sym}}$ allows to replace the remaining phase factors with `fftshift`. The result can be simplified to only require `fftshift` `ift` once. Since it is then only needed in the construction of the frequency space coordinates, many FFT libraries provide a helper function called `fftfreq` to construct them directly:

$$\text{Set} \quad f_{\min} := f_{\min}^{\text{sym}}$$

$$\Rightarrow \frac{d}{dx} g_n = e^{+2\pi i \, f_{\min}^{\text{sym}} \, n\Delta x} \, \texttt{ifft}_m($$

$$2\pi i \left( f_{\min}^{\text{sym}} + m\Delta f \right) \texttt{fft}_m \left( g_n \, e^{-2\pi i \, f_{\min}^{\text{sym}} \, n\Delta x} \right)$$

$$)$$

$$= \texttt{ifft}_m(\texttt{ifftshift}_m($$

$$2\pi i \left( f_{\min}^{\text{sym}} + m\Delta f \right) \texttt{fftshift}_m(\texttt{fft}_m(g_n))$$

$$))$$

$$= \texttt{ifft}_m($$

$$\texttt{ifftshift}_m \left( 2\pi i \left( f_{\min}^{\text{sym}} + m\Delta f \right) \right)$$

$$\texttt{ifftshift}_m(\texttt{fftshift}_m(\texttt{fft}_m(g_n)))$$

$$)$$

$$= \texttt{ifft}_m($$

$$2\pi i \, \texttt{ifftshift}_m \left( f_{\min}^{\text{sym}} + m\Delta f \right) \texttt{fft}_m(g_n)$$

$$)$$

$$= \texttt{ifft}_m($$

$$2\pi i \, \texttt{fftfreq}_m(N, \Delta x) \, \texttt{fft}_m(g_n)$$

$$).$$

As shown in the examples above, the specific possible optimizations differ a lot for different use cases.

## 3 The FFTArray Library

FFTArray enables an easy to use general discretized Fourier transform while comprehensively addressing all special cases outlined in section 2.3 by implementing the general discretized Fourier transforms (gd(I)FT) in eqs. (16) and (19). By providing a modular toolkit for seamlessly and efficiently manipulating discretized functions in their position and frequency space representations it replaces the handling of grid-specific complexities with the common FFT library helpers `fftshift`, `ifftshift` and `fftfreq`. In contrast to the few discrete shifts supported by these methods FFTArray supports arbitrarily shifted coordinate grids.

Two core classes, `Dimension` and `Array`, automatically track coordinate grids and apply FFTs and phase factors where necessary. The `Dimension` class (section 3.1) encapsulates the position and frequency grids of a single dimension. The parameters of both domains can be initialized in the way most convenient for the current problem via a constraint solver. It ensures that the constraint in eq. (11) is always fulfilled and $N$ is even or a power of two. The `Array` class (section 3.2) manages multi-dimensional sampled functions, their Fourier transforms and general mathematical operations. It stores the sample values and `Dimension` objects, while tracking for each axis whether it is in position space ($g_n$) or frequency space ($G_m$, as formalized in section 2). Transformations between domains are executed implicitly by setting the desired space for each dimension and automatically handling the required parts of the gd(I)FT. During arithmetic operations involving `Array` instances dimensions are broadcast based on their names. Both classes are immutable and all operations on an `Array` create a new array which reuses the values of one of its inputs if possible.

To minimize computational overhead, unnecessary scale and phase factors are automatically omitted in a user-controllable and predictable manner as detailed in section 3.3. By leveraging the Python Array API standard [21] (cf. section 3.4), FFTArray ensures portability and allows for higher performance by utilizing hardware accelerators like GPUs.

Smaller code examples are given in a Read-eval-print-loop (REPL) style. Python expressions and code lines are preceded with a >>> and the result of that line is printed below[2]. Variables are prefixed with `np_` for NumPy Arrays, `dim_` for `fa.Dimension` objects and `arr_` for `fa.Array` objects. The import below is used in all examples in this chapter.

```
1  >>> import fftarray as fa
```

Additionally variables defined in earlier snippets are still available for reuse in later snippets.

## 3.1 The `Dimension` class: Defining Coordinate Grids

FFTArray automatically handles arbitrary coordinate grids and ensures their validity such that the user can simply choose the grid best suited to the problem. This design avoids mistakes when defining grids which must fulfill the dependencies given in table 1. Additionally it standardizes which variables are used to uniquely define the grid coordinates.

The `Dimension` class represents the coordinate grids for one dimension in position and frequency space. It stores the name of the dimension and the numerical parameters $N$, $\Delta x$, $x_{\min}$, and $f_{\min}$. $\Delta f$ can be obtained via eq. (11) from a given $N$ and $\Delta x$. To initialize a `Dimension` from a set of parameters which does not consist of exactly $N$, $\Delta x$, $x_{\min}$, and $f_{\min}$, the system of equations in table 1 must be solved. For example, given the grid spacings $\Delta x$ and $\Delta f$ one would have to solve eq. (11) for $N$. However, this solution for $N$ might not be an integer, which means, that either $\Delta f$ or $\Delta x$ or both parameters need to be adapted. Moreover, specific applications may have their own particular set of constraints on the definition on the grids, e.g., the position grid may need to be symmetric around a center point $x_{\text{middle}}$ or the frequency grid may need to accommodate a function with an upper band limit of $f_{\max}$.

FFTArray allows to use any combination of grid parameters to initialize a `Dimension`. Internally, FFTArray uses the z3 solver [22] for solving the constraints in table 1. This way, the user is not required to do so by hand. For example, a grid with $N = 2048$, $x_{\min} = -100$, $x_{max} = 50$ that is centered in frequency space can be initialized with[3]:

```
1  >>> fa.dim_from_constraints("x", n=2048, pos_min=-100., pos_max=50.,
   ↪ freq_middle=0.)
2  Dimension(name='x', n=2048, d_pos=0.0733, pos_min=-100.0, freq_min=-6.823)
```

The user defined parameter set may not correspond to a uniquely solvable set of equations. In such cases, FFTArray guides the user to a working solution via its error messages. If the given constraints allow many different solutions, a `NoUniqueSolutionError` is thrown, suggesting additional parameters which could complete the set of constraints leading to a unique solution. If, on the other hand, the given constraints have no solution at all, a `NoSolutionFoundError` suggests parameters to be removed. The solver also supports cases where an exact solution would require a non-integer number of grid points. In this case the error message suggests which parameters could be marked as `loose_params` to be automatically adapted. These

---

[2]Whether an output is printed as well as the shown output are not necessarily identical to what one would see in the real CPython interpreter REPL. We sometimes added the output of the assigned value which is normally not printed, switched the `__repr__` with the `__str__` representation and shortened floats in order to aid understanding.

[3]The REPL outputs in this section omit the `dynamically_traced_coords=False` member of the `Dimension` class and round floats to more readable lengths for pedagogical reasons.

| Math | Name in Code | Description |
|------|--------------|-------------|
| $1 = N \Delta f \Delta x, \quad N \in \mathbb{N}_+$ | `n` | Number of grid points |
| $\Delta x$ | `d_pos` | Spacing between two grid points in position space. |
| $\Delta f$ | `d_freq` | Spacing between two grid points in frequency space. If the unit of position space is $[x]$, the unit in frequency space is its inverse $[f] = [x]^{-1}$. This is a rotational frequency in cycles as opposed to an angular frequency. |
| $x_{\min}$ | `pos_min` | The smallest position grid point. |
| $x_{\max} = x_{\min} + (N-1) \Delta x$ | `pos_max` | The largest position grid point. |
| $x_{\text{middle}} = \begin{cases} 0.5\,(x_{\min} + x_{\max} + \Delta x), \\ N \text{ even} \\ 0.5\,(x_{\min} + x_{\max}), \\ N \text{ odd} \end{cases}$ | `pos_middle` | The middle of the position grid. |
| $x_{\text{extent}} = x_{\max} - x_{\min}$ | `pos_extent` | The length of the position grid. Note that this is one $\Delta x$ smaller than the period $x_{\text{period}}$. |
| $f_{\min}$ | `freq_min` | The smallest frequency grid point. |
| $f_{\max} = f_{\min} + (N-1) \Delta f$ | `freq_max` | The largest frequency grid point. |
| $f_{\text{middle}} = \begin{cases} 0.5\,(f_{\min} + f_{\max} + \Delta f), \\ N \text{ even} \\ 0.5\,(f_{\min} + f_{\max}), \\ N \text{ odd} \end{cases}$ | `freq_middle` | The middle of the frequency grid. |
| $f_{\text{extent}} = f_{\max} - f_{\min}$ | `freq_extent` | The length of the frequency grid. Note that this is one $\Delta f$ smaller than the period $f_{\text{period}}$. |

Table 1: All parameters of the constraint system between position and frequency space. Math is the naming in section 2 while Code lists the naming adopted in the actual source code. Those names where chosen such that they are independent of the name of the used dimension to properly support multi-dimensional use-cases.

parameters are then adapted such that $N$ is even or a power of two. Rounding up means that the extent of a space is always increased and the spacing of samples decreased. Below we give an example, where $\Delta x$ has been decreased such that it fits the constraints of $\Delta x \leq 0.1$ and $N$ being a power of two:

```
1  >>> fa.dim_from_constraints("x", d_pos=0.1, d_freq=0.05, pos_min=-9.,
   ↪  freq_min=-6.4, n="power_of_two", loose_params=["d_pos"])
2  Dimension(name='x', n=256, d_pos=0.078125, pos_min=-9.0, freq_min=-6.4)
```

Finally, the grid coordinates can be output as NumPy arrays with `dim.values(space)` or directly packed into a new `Array` via `fa.coords_from_dim(dim, space)` (cf. section 3.2.1).

## 3.2   The `Array` class: Managing Values in Position and Frequency Space

The `Array` class streamlines the handling of the gd(I)FT by automatically managing dimension-wise scale and phase factors in both, position and frequency space. This eliminates the need for the user to manually track coordinate grid compatibility and apply factors when combining multidimensional arrays with each other.

In contrast to a naive implementation like functions acting directly on NumPy arrays, which would redundantly apply scale and phase factors even when unnecessary, the Array class avoids such inefficiencies by tracking the current space of each Dimension and enabling transformations between spaces (see section 3.2.2). It automatically applies the correct factors only when required and elides them during arithmetic operations where possible (see sections 3.2.3 and 3.3). It does so by encapsulating the samples of a multidimensional function together with each `Dimension`. Correct dimension broadcasting and tracking of `Dimension` objects is enabled by associating each dimension with a unique name, similar to Unidata's self-describing Common Data Model, netCDF and xarray [23–25].

As outlined in section 3.4, the operations of the Array class are built upon the Python Array API [21] to enable portability with different array libraries and support computations on GPUs.

### 3.2.1   Initialization

The `Array` class handles the gd(I)FT by storing the function values, the `Dimension` object and the current space for each dimension. It can be initialized in various ways depending on the use case; here we highlight the two most important ones. For a complete list of initialization functions we refer to the API reference in the documentation [26]. The most common way to initialize an `Array` is to directly fill it with the coordinate values of a `Dimension`:

```
1   >>> dim_x: fa.Dimension = fa.dim("x", pos_min=-0.1, freq_min=0., d_pos=0.2, n=4)
2   Dimension(name='x', n=4, d_pos=0.2, pos_min=-0.1, freq_min=0.0)
3   # convert the Dimension into an Array with values given by the coordinate grid:
    ↪  g(x) = x
4   >>> arr_x: fa.Array = fa.coords_from_dim(dim_x, "pos")
5   <fftarray.Array (x: 2^2)> Size: 32 bytes
6   |dimension | space |    d     |   min    |  middle  |   max    |  extent  |
7   +----------+-------+----------+----------+----------+----------+----------+
8   |    x     |  pos  |   0.20   |  -0.10   |   0.30   |   0.50   |   0.60   |
9   Values<array_api_compat.numpy>:
10  [-0.1  0.1  0.3  0.5]
```

After its initialization, the `Array` can be used in any arithmetic expression like for example `x**2`. Alternatively, one can wrap a preexisting bare array via `fa.array` with Dimension objects and define its current space:

```
1  >>> import numpy as np
2  >>> dim_x = fa.dim("x", pos_min=-0.1, freq_min=0., d_pos=0.2, n=4)
3  Dimension(name='x', n=4, d_pos=0.2, pos_min=-0.1, freq_min=0.0)
4  >>> np_values = np.array([5.,6.,7.,8.]) # correspond to the coordinate defined by
   ↪ dims
5  >>> arr_pos = fa.array(np_values, [dim_x], "pos") # 1d Array in position space
6  <fftarray.Array (x: 2^2)> Size: 32 bytes
7  |dimension | space |    d     |   min    | middle   |   max    |  extent  |
8  +----------+-------+----------+----------+----------+----------+----------+
9  |    x     |  pos  |   0.20   |  -0.10   |   0.30   |   0.50   |   0.60   |
10 Values<array_api_compat.numpy>:
11 [5. 6. 7. 8.]
```

Instances of `Array` can hold all data types of the Python Array API Standard 2024.12 [21].

### 3.2.2 Fourier Transforms

A key design goal of FFTArray is to make the user deliberately select the required position or frequency space representation rather than explicitly executing transforms. The space for each dimension can be set individually and will trigger a gd(I)FT on the internal values. Thereby, the code automatically documents clearly which representation is used for each operation while the actual execution of the gd(I)FT becomes implicit and can be skipped if it is unnecessary.

```
1  arr_freq = arr_pos.into_space("freq") # change space: "pos" -> "freq"
2  arr_pos = arr_freq.into_space("pos") # change space: "freq" -> "pos"
3  arr_pos = arr_pos.into_space("pos") # No operation done because unnecessary.
```

Changing the space is only possible on a floating point `Array`. When performing space-changing operations, a real valued `Array` is automatically upcast to a complex floating point format of the same precision. It is also possible to set different spaces per dimension. This can be useful when mixing dimensions like time and space within the same array, which are usually not transformed simultaneously.

### 3.2.3 Arithmetic Operations and Broadcasting

FFTArray enables writing down most computations very similarly to their analytic counterparts. The `fftarray` namespace contains all element-wise functions of the Python Array API standard [21] like `sin`, `multiply`, etc. These implementations of arithmetic operations are also used to support all common element-wise unary and binary Python operators between `Array` instances as well as with scalars. Statistical functions like `sum` or `max` work with dimension names instead of axes indices and `integrate` additionally uses the spacings $\Delta x$ and $\Delta f$ stored in `Dimension` as integration elements.

When combining multiple arrays in an arithmetic operation their values and metadata are automatically aligned and broadcast by their `Dimension` name. The coordinate grids and space of equally named dimensions must exactly match between all operands. Because the results of an arithmetic operation differ when done in a different space there is no automatic conversion between position and frequency space.

The following example showcases a combination of the aforementioned features to define a two-dimensional Gaussian function:

```
1  >>> dim_x = fa.dim_from_constraints("x", pos_min=-1., pos_max=0., n=2,
   ↪ freq_middle=0.)
2  >>> dim_y = fa.dim_from_constraints("y", pos_min=-2., pos_max=1., n=4,
   ↪ freq_middle=0.)
```

```
3  >>> arr_x = fa.coords_from_dim(dim_x, "pos")
4  >>> arr_y = fa.coords_from_dim(dim_y, "pos")
5  >>> arr_gauss_2d = fa.exp(-(arr_x**2 + arr_y**2)/0.2) # same width along x and y,
   ↪  centered around (x,y)=(0,0)
6  <fftarray.Array (x: 2^1, y: 2^2)> Size: 64 bytes
7  |dimension | space |    d    |   min    | middle  |   max    |  extent  |
8  +----------+-------+---------+---------+---------+---------+---------+
9  |    x     |  pos  |  1.00   |  -1.00  | 0.00e+00 | 0.00e+00 |   1.00   |
10 |    y     |  pos  |  1.00   |  -2.00  | 0.00e+00 |  1.00   |   3.00   |
11 Values<array_api_compat.numpy>:
12 [[1.389e-11 4.540e-05 6.738e-03 4.540e-05]
13  [2.061e-09 6.738e-03 1.000e+00 6.738e-03]]
```

Note that for all operations, the `Dimension` objects are properly stored in the resulting `Array` so that the Gaussian array can be easily transformed from position into frequency space with `arr_gauss_2d.into_space("freq")`.

### 3.2.4 Indexing

Indexing is supported via index or coordinate in any of the dimensions, similar to xarray.

```
1  # Select the point in the middle of both x and y direction.
2  >>> arr_gauss_2d.sel({"x": dim_x.pos_middle, "y": dim_y.pos_middle},
   ↪  method="nearest")
3  <fftarray.Array (x: 2^0, y: 2^0)> Size: 8 bytes
4  |dimension | space |    d    |   min    | middle  |   max    |  extent  |
5  +----------+-------+---------+---------+---------+---------+---------+
6  |    x     |  pos  |  1.00   | 0.00e+00 | 0.00e+00 | 0.00e+00 | 0.00e+00 |
7  |    y     |  pos  |  1.00   | 0.00e+00 | 0.00e+00 | 0.00e+00 | 0.00e+00 |
8  Values<array_api_compat.numpy>:
9  [[1.]]
```

When slicing, the `Dimensions` are sliced as well and automatically used in the resulting `Array`. Slicing off a part of position space increases the value of `d_freq`, and conversely, slicing off a part of frequency space increases the value of `d_pos` due to their reciprocal relationship in eq. (11).

```
1  # Select the first 3 points in y dimension.
2  # The Dimension object is automatically adjusted to correctly cover the selected
   ↪  points.
3  >>> arr_gauss_2d.isel({"y": slice(0,3)})
4  <fftarray.Array (x: 2^1, y: 3)> Size: 48 bytes
5  |dimension | space |    d    |   min    | middle  |   max    |  extent  |
6  +----------+-------+---------+---------+---------+---------+---------+
7  |    x     |  pos  |  1.00   |  -1.00  | 0.00e+00 | 0.00e+00 |   1.00   |
8  |    y     |  pos  |  1.00   |  -2.00  |  -1.00  | 0.00e+00 |   2.00   |
9  Values<array_api_compat.numpy>:
10 [[1.389e-11 4.540e-05 6.738e-03]
11  [2.061e-09 6.738e-03 1.000e+00]]
```

Indexing with sub steps is not supported because the domain of the other space could be adjusted in multiple ways. Increasing the step size in one space reduces the extent of the other space. This extent reduction could be done on either end of the space or on both ends in different ways. Since there is no sensible default for this reduction, sub steps are not supported.

### 3.3 Lazy Phase Factor Application

One of the major design goals for FFTArray is to achieve high computational performance by avoiding unnecessary computations, particularly the application of scale and phase factors during gd(I)FTs. FFTArray can skip applying these factors if they would cancel out during back-and-forth transformations. The user-accessible values are always given in the representation with all phase and scale factors applied. Skipping certain factors in-between operations avoids floating-point inaccuracies which could otherwise accumulate if the factors were actually applied and reversed on each transform. Users retain control over this behavior as part of the programming model: if desired, they can explicitly influence or disable it. This explicit control ensures predictable outcomes, as opposed to relying on automatic optimizations, which may vary unpredictably across program versions or configurations.

In order to automate the scale and phase factor application while still ensuring user control, we introduce the concept of "lazy application". After performing an (i)FFT using `into_spa`⌋ `ce`, the scale and phase factors are not directly applied but instead, the resulting `Array` tracks that they are missing via the flag `factors_applied=`False. This flag is separate for each dimension. All operations on `Array` objects in FFTArray take `factors_applied` into account to ensure the correct result and avoid applying these factors if possible.

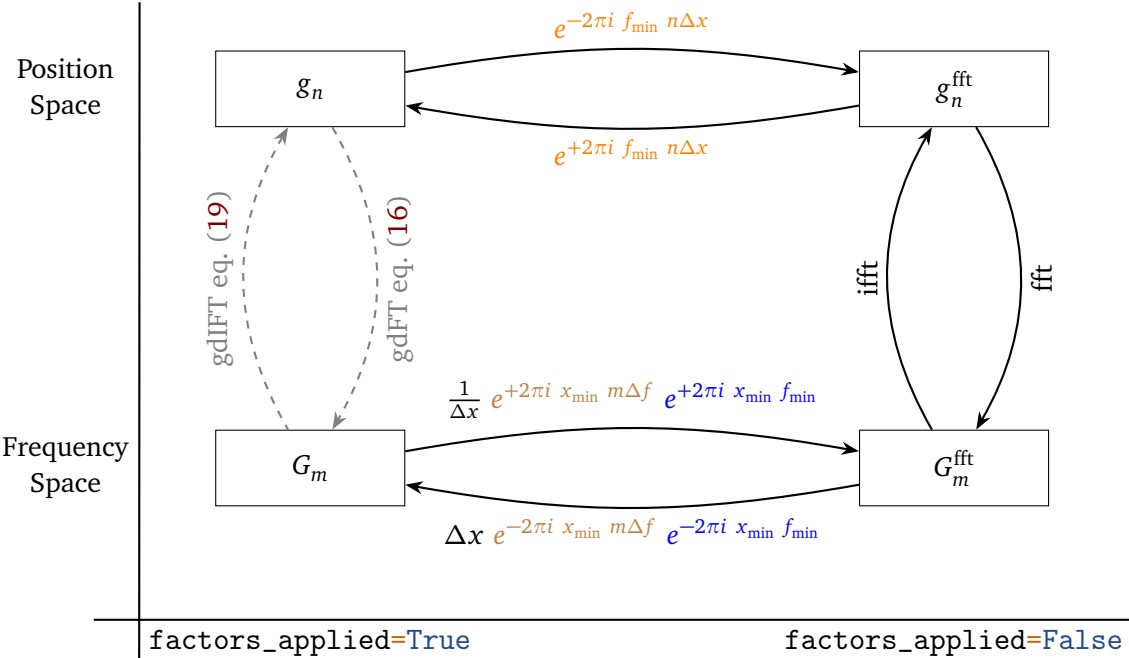

Figure 2: The four different internal states for the values of an FFTArray. By default (`eager=`False) each operation on the FFTArray minimizes the number of state transitions.

The flag `factors_applied` implements a new set of states of `Array`s internal values on top of the current `space` flag, namely, $g_n^{\text{fft}}$ and $G_m^{\text{fft}}$. The new states are required directly before computing the (I)FFT and are derived from eqs. (16) and (19) with the colored phase factors being the same. They are called $g_n^{\text{fft}}$ and $G_m^{\text{fft}}$:

$$g_n^{\text{fft}} := g_n \, e^{-2\pi i \, f_{\min} \, n\Delta x}, \tag{28}$$

$$G_m^{\text{fft}} := G_m \, e^{+2\pi i \, x_{\min} \, m\Delta f} \, e^{+2\pi i \, x_{\min} \, f_{\min}} / \Delta x. \tag{29}$$

With these definitions, the gd(I)FT from $g_n$ to $G_m$ or vice versa can be split into three separate steps as depicted in fig. 2. For the gdFT, the first step is multiplying the position space values

$g_n$ with $e^{-2\pi i\ f_{\min}\ n\Delta x}$, which results in $g_n^{\text{fft}}$. The second step is performing the FFT on $g_n^{\text{fft}}$ to compute $G_m^{\text{fft}}$. In the third step, we multiply $G_m^{\text{fft}}$ with $e^{+2\pi i\ x_{\min}\ m\Delta f}\ e^{+2\pi i\ x_{\min}\ f_{\min}}/\Delta x$ to get the actual frequency space values $G_m$. The advantage of splitting this process into three individual steps is that the final transitions $g_n^{\text{fft}}$ to $g_n$ ($G_m^{\text{fft}}$ to $G_m$) to the states with all phase factors applied can be skipped or simplified for many operations as shown in sections 3.3.1 to 3.3.4.

By default, the lazy evaluation minimizes state transitions in `factors_applied` in fig. 2 for all `Array` operations. The user can deactivate this behavior by setting the attribute `eag⌋er: Tuple[bool]` to `True` for each dimension. In this case, the phase and scale factors are applied directly after each space change. Combining two `Array` instances raises an error if their `eager` attributes do not match.

All operations behave, up to numerical accuracy, as if they had been executed with the phase and scale factors applied. An `Array` object (`arr`) only allows public access to $g_n$ (`arr⌋.values("pos")`) and $G_m$ (`arr.values("freq")`). In cases in which multiple calculations require the factors to be applied to the same `Array` it can improve performance to explicitly change the value of `factors_applied` with `arr = arr.into_factors_applied(True)`.

In order to take advantage of the lazy phase factor application, the functions `abs`, add, `subtract`, `multiply` and `divide` use optimized code paths. These optimized implementations apply to both the free-standing functions in the `fftarray` name space as well as their unary and binary counterparts on the `Array` class. Apart from `abs`, the above functions require two `Array` objects $g_n$ and $h_n$ as arguments and thus need separate rules for each combination of `factors_applied`. This chapter only focuses on a single dimension since phase and scale factor application is independent for each dimension. Additionally it only focuses on position space, since except for `abs` the same rules apply to frequency space just with the other phase and scale factors from eqs. (28) and (29).

The values stored internally in an `Array` are given by

$$g_n^{\text{int}}(s) := \begin{cases} g_n & \text{if } \texttt{factors\_applied=True} \\ g_n^{\text{fft}} & \text{if } \texttt{factors\_applied=False} \end{cases} = g_n \left[ e^{-2\pi i\ f_{\min}\ n\Delta x} \right]^s, \qquad (30)$$

using 28. In order to be able to write `factors_applied` in analytical expressions, we encode it into a variable $s$:

$$s := \begin{cases} 0 & \text{if } \texttt{factors\_applied=True} \\ 1 & \text{if } \texttt{factors\_applied=False} \end{cases}. \qquad (31)$$

The correct full representation is obtained via moving the exponential to the other side:

$$g_n = g_n^{\text{int}}(s) \left[ e^{+2\pi i\ f_{\min}\ n\Delta x} \right]^s. \qquad (32)$$

### 3.3.1 Addition and subtraction

Addition has two input arrays with user-facing values $g_n$ and $h_n$ which should be added to get the result $f_n := g_n + h_n$. The derivation of the optimized rules for addition starts in terms of the two internal Array values $g_n^{\text{int}}(s_1)$ and $h_n^{\text{int}}(s_2)$ with their respective `factors_applied` states $s_1$ and $s_2$. It is not always possible to completely avoid the application of scale and phase factors and still get a result of the form $f_n^{\text{int}}(s^{\text{res}})$ with $s^{\text{res}} \in \{0, 1\}$. Therefore we introduce possible adjustments to each input as $s_1^{\text{op}}, s_2^{\text{op}} \in \{-1, 0, 1\}$. These adjustments can be used to switch the representation of the values in the array object between $g_n$ and $g_n^{\text{fft}}$ before the

| eager | $s_1$ | $s_2$ | $s_1^{\text{op}}$ | $s_2^{\text{op}}$ | $s^{\text{res}} = s_1 + s_1^{\text{op}} = s_2 + s_2^{\text{op}}$ |
|---|---|---|---|---|---|
| False | 1 | 1 | 0 | 0 | 1 |
| False | 1 | 0 | 0 | 1 | 1 |
| False | 0 | 1 | 1 | 0 | 1 |
| False | 0 | 0 | 0 | 0 | 0 |
| True | 1 | 1 | 0 | 0 | 1 |
| True | 1 | 0 | -1 | 0 | 0 |
| True | 0 | 1 | 0 | -1 | 0 |
| True | 0 | 0 | 0 | 0 | 0 |

Table 2: Look-up-table for `add` and `subtract`. It encodes which inputs need phase and scale factors applied for each dimension. If possible the factors are factored out. `eager` acts as a tie breaker when any of the two inputs could be adjusted in order to get a correct result.

operation. For addition this results in:

$$
g_n^{\text{int}}(s_1)\left[e^{-2\pi i\,f_{\min}\,n\Delta x}\right]^{s_1^{\text{op}}} + h_n^{\text{int}}(s_2)\left[e^{-2\pi i\,f_{\min}\,n\Delta x}\right]^{s_2^{\text{op}}}
$$
$$
= g_n\left[e^{-2\pi i\,f_{\min}\,n\Delta x}\right]^{s_1+s_1^{\text{op}}} + h_n\left[e^{-2\pi i\,f_{\min}\,n\Delta x}\right]^{s_2+s_2^{\text{op}}} \tag{33}
$$
$$
\overset{!}{=} f_n\left[e^{-2\pi i\,f_{\min}\,n\Delta x}\right]^{s^{\text{res}}}, \quad f_n := g_n + h_n \tag{34}
$$
$$
\Rightarrow s^{\text{res}} \overset{!}{=} s_1 + s_1^{\text{op}} = s_2 + s_2^{\text{op}}. \tag{35}
$$

Now we need to solve eq. (35) while keeping $s_1^{\text{op}} = 0$ and $s_2^{\text{op}} = 0$ as much as possible in order to avoid having to apply phase and scale factors to the input operands. In the case of $s_1 = s_2$ this is directly possible with $s_1^{\text{op}} = s_2^{\text{op}} = 0$ leading to $s^{\text{res}} = s_1 = s_2$. If $s_1 \neq s_2$, one of the two input `Arrays` needs to be adjusted, so either $s_1^{\text{op}} \neq 0$ or $s_2^{\text{op}} \neq 0$. Since the choice whether to adjust $s_1$ or $s_2$ has no performance implications, the `eager` attribute acts as a tie breaker. If `eager=False`, $s_1^{\text{op}}$ and $s_2^{\text{op}}$ are chosen such that $s^{\text{res}} = 1$ which corresponds to `factors_applied=False`. If `eager=True`[4], the $s_1^{\text{op}}$ and $s_2^{\text{op}}$ are chosen such that $s^{\text{res}} = 0$ and therefore the resulting array will have `factors_applied=True`. The logic outlined above is encoded in table 2. These lookup tables are also used to implement that logic for each operation in the actual library. In each operation they are consulted for each dimension separately. The above derivation can be done identically for subtraction.

### 3.3.2 Multiplication

In the case of `multiply` the commutativity of complex multiplication can be exploited. The multiplication of two arrays $g_n$ and $h_n$ can be written as

---

[4]In this case `factors_applied=False` is pretty uncommon because it requires that the user manually changed either the `factors_applied` or `eager` attribute. But for example manually setting `factors_applied=False` on an `Array` with `eager=True` is a valid operation.

| eager | $s_1$ | $s_2$ | $s_1^{\text{op}}$ | $s_2^{\text{op}}$ | $s^{\text{res}} = s_1 + s_1^{\text{op}} + s_2 + s_2^{\text{op}}$ |
|---|---|---|---|---|---|
| False/True | 1 | 1 | 0 | -1 | 1 |
| False/True | 1 | 0 | 0 | 0 | 1 |
| False/True | 0 | 1 | 0 | 0 | 1 |
| False/True | 0 | 0 | 0 | 0 | 0 |

Table 3: Look-up-table for `multiply`. It encodes which inputs need phase and scale factors applied for each dimension. Since the multiplication of the inputs commutes with the factors, they can be propagated through without applying them except in the case of two phase factors which require the application of one of them.

$$g_n^{\text{int}}(s_1)\left[e^{-2\pi i\, f_{\min}\, n\Delta x}\right]^{s_1^{\text{op}}} \times h_n^{\text{int}}(s_2)\left[e^{-2\pi i\, f_{\min}\, n\Delta x}\right]^{s_2^{\text{op}}} \tag{36}$$

$$= g_n\left[e^{-2\pi i\, f_{\min}\, n\Delta x}\right]^{s_1+s_1^{\text{op}}} \times h_n\left[e^{-2\pi i\, f_{\min}\, n\Delta x}\right]^{s_2+s_2^{\text{op}}}$$

$$= (g_n \times h_n)\left[e^{-2\pi i\, f_{\min}\, n\Delta x}\right]^{s_1+s_1^{\text{op}}+s_2+s_2^{\text{op}}} \tag{37}$$

$$\overset{!}{=} f_n\left[e^{-2\pi i\, f_{\min}\, n\Delta x}\right]^{s^{\text{res}}}, \quad f_n := g_n \times h_n \tag{38}$$

$$\Rightarrow s^{\text{res}} \overset{!}{=} s_1 + s_1^{\text{op}} + s_2 + s_2^{\text{op}}. \tag{39}$$

Table 3 solves the resulting eq. (39) such that it minimizes the number of entries where $s_j^{\text{op}} \neq 0$ and therefore also minimizes the amount of additional arithmetic. Only the case of $s_1 = s_2 = 1$ requires the application of additional phase factors, which we have arbitrarily chosen to apply to the second `Array`.

### 3.3.3 Division

Division is similar to multiplication but with the difference that the signs of $s_2$ and $s_2^{\text{op}}$ in the equality condition for $s^{\text{res}}$ are flipped:

$$\frac{g_n^{\text{int}}(s_1)\left[e^{-2\pi i\, f_{\min}\, n\Delta x}\right]^{s_1^{\text{op}}}}{h_n^{\text{int}}(s_2)\left[e^{-2\pi i\, f_{\min}\, n\Delta x}\right]^{s_2^{\text{op}}}} = \frac{g_n\left[e^{-2\pi i\, f_{\min}\, n\Delta x}\right]^{s_1+s_1^{\text{op}}}}{h_n\left[e^{-2\pi i\, f_{\min}\, n\Delta x}\right]^{s_2+s_2^{\text{op}}}} \tag{40}$$

$$= \frac{g_n}{h_n}\left[e^{-2\pi i\, f_{\min}\, n\Delta x}\right]^{s_1+s_1^{\text{op}}-s_2-s_2^{\text{op}}} \tag{41}$$

$$\overset{!}{=} f_n\left[e^{-2\pi i\, f_{\min}\, n\Delta x}\right]^{s^{\text{res}}}, \quad f_n := \frac{g_n}{h_n} \tag{42}$$

$$\Rightarrow s^{\text{res}} := s_1 + s_1^{\text{op}} - s_2 - s_2^{\text{op}}. \tag{43}$$

Solving eq. (43) in table 4 also requires an extra phase factor in only in one case. Again the operand can be chosen arbitrarily, though one needs to take care to use the correct sign.

### 3.3.4 Absolute values

`abs(x)` removes the phase of a complex number. Therefore, any not yet-applied phase-factors can simply be dropped and the result will always be with `factors_applied=True`. If the values are in frequency space and `factors_applied=False`, the scale factor $[\Delta f N]^s$ needs to be applied before or after computing the absolute value of the internal values.

The frequency space scale factors $[\Delta f N]^s$ are computed after the `abs` operation, because it is more efficient to apply them to a real-valued array instead of a complex-valued array.

| eager | $s_1$ | $s_2$ | $s_1^{\mathrm{op}}$ | $s_2^{\mathrm{op}}$ | $s^{\mathrm{res}} = s_1 + s_1^{\mathrm{op}} - s_2 - s_2^{\mathrm{op}}$ |
|---|---|---|---|---|---|
| False/True | 1 | 1 | 0 | 0 | 0 |
| False/True | 1 | 0 | 0 | 0 | 1 |
| False/True | 0 | 1 | 0 | -1 | 0 |
| False/True | 0 | 0 | 0 | 0 | 0 |

Table 4: Look-up-table for `divide`. It encodes which inputs need phase and scale factors applied for each dimension. Compared to `multiply` the signs of the second operand are flipped but still only one case needs an actual correction to implement the operation correctly.

### 3.3.5  Showcase

The logic described in this section can be used to elide the application of phase and scale factors in a wide class of algorithms, which we demonstrate in section 4. Below we showcase a compact implementation of these optimizations.

```python
import fftarray as fa

# Compute the dimension properties.
dim_x = fa.dim_from_constraints("x", pos_min=-1., pos_max=1., n=1024,
    freq_middle=0.)

# Initialize the coordinate grids in position and frequency space.
# Those are real-valued and therefore have to have factors_applied=True.
# They default to eager=False.
arr_x = fa.coords_from_dim(dim_x, "pos") # g_n
arr_f = fa.coords_from_dim(dim_x, "freq") # G_m

# The result of the square with factors_applied=True is again factors_applied=True
arr_pos1 = arr_x**2 # g_n
# Changing the space leaves the array with factors_applied=False. The factors have
    not been applied yet.
arr_freq1 = arr_pos1.into_space("freq") # G_m^fft

arr_freq2 = arr_freq1 * arr_f # G_m^fft, multiplication of factors_applied=False and
    factors_applied=True leads to factors_applied=False, no factors are actually
    applied during this operation.

# Because arr_freq2 is in the fft representation (G^fft_m) the ifft can be applied
    directly.
# Therefore this is only a call to ifft, no factors in frequency or position space
    necessary before the transformation.
arr_pos2 = arr_freq2.into_space("pos") # g_n^fft
# eager=False acts as a tie breaker, so the result has factors_applied=False.
arr_freq3 = arr_freq2 + 5 # G_m^fft, if eager=True it would be G_m

arr_freq3 = fa.exp(arr_freq2) # G_m, factors applied before expontential function
np_arr_freq2 = arr_freq2.values("freq") # values of G_m in a plain NumPy array

arr_freq4 = fa.abs(arr_freq2) # G_m, only scaling factors were applied since
    |G_m| = Δx|G_m^fft|
```

## 3.4  Python Array API

FFTArray is built on top of the Python Array API to leverage the speed-ups offered by modern hardware accelerators like GPUs. The specific needs of hardware accelerators for deep learning

and scientific compute led in the last years to the creation of multiple new python libraries for array computing. Each of these libraries has different trade-offs. NumPy is almost universally available in the Python ecosystem and has low start-up and per operation overhead. JAX and PyTorch both enable significant speed-ups on GPUs but have different designs and methods to translate a Python program to run on a GPU. A library like FFTArray is in principle agnostic to these details and could be built on top of any of these libraries. But the different trade-offs and histories of these libraries cause them to have different application programming interfaces (APIs). To enable a library like FFTArray to take advantage of all of these libraries from a single source, the Python Array API standard was created. It is developed by the Consortium for Python Data API Standards [21] and defines a common minimal set of functionality. Adoption of this standard is facilitated by the `array-api-compat` library [27]. It provides a wrapper over libraries like NumPy, PyTorch and JAX to fix any standard-violating behavior of the individual array libraries.

All array operations in FFTArray, from basic arithmetic to (i)FFTs, are forwarded to the underlying library via `array-api-compat`. Every Array API compliant library provides a namespace which we will call xp. This namespace exposes at least a standardised set of functionality like `xp.sin` or `xp.fft.fftn`. Every arithmetic operation on an `Array` from direct additions to functions like `fa.sin` are automatically dispatched to the functions of the Array API namespace xp of the array. As an example `fa.sin(arr).values("pos")` is equivalent to `xp.sin(arr.values("pos"))`. This also means that `array-api-compat` and the wrapped array library define any not standardized behavior of FFTArray. A notable example for not standardized behavior which is commonly used are the generally more relaxed type promotion rules. For example `np.asarray(True)+2` results in 3 with NumPy although this upcasting behavior (`bool` to `int`) is not guaranteed by the standard. When using NumPy as the backend for FFTArray this upcast is performed while with other backends it might not.

The array library can be different for each individual `Array`. An `Array` can be initialized with values from any Array API compatible library, e.g., `np.ndarray`. The Array API namespace is then automatically deduced. The other array creation functions can optionally be given an Array API namespace. If it is not possible to determine the used array library from the input like in the case of a list, FFTArray uses a user-configurable default namespace which itself defaults to NumPy. The Array API namespace of any `Array` can also always be inspected via `arr.xp` and changed via `new_arr = arr.into_xp(xp)`. This conversion behavior currently always goes through a NumPy array and only supports explicitly implemented libraries because it is not covered by the standard at the moment.

If the user attempts to mix multiple different namespaces, an error is thrown because it is unclear in which namespace the operation should be executed. Therefore, the user needs to ensure that all arrays which are combined in an operation use the same underlying array library. In the example below, we show how to set it to the `jax.numpy` namespace. In this case, any operations on the arrays `arr_g_x_jax` or `arr_lin_jax` are executed by JAX.

```
1  >>> import jax.numpy as jnp
2  >>> import fftarray as fa
3  >>> dim_x = fa.dim_from_constraints("x", pos_min=-1., pos_max=1., n=4,
   ↪  freq_middle=0.)
4
5  >>> arr_g_x_jax = fa.coords_from_dim(dim_x, "pos", xp=jnp) # set namespace
   ↪  explicitly to jax.numpy
6  <fftarray.Array (x: 2^2)> Size: 16 bytes
7  |dimension | space |   d    |   min    | middle  |   max    |  extent  |
8  +----------+-------+--------+----------+---------+----------+----------+
9  |    x     |  pos  |  0.67  |  -1.00   |  0.33   |  1.00    |  2.00    |
10 Values<jax.numpy>:
11 [-1.        -0.3333333   0.33333337  1.        ]
```

```
12
13   >>> arr_lin_jax = fa.array(jnp.linspace(0., 1.5, 4), dim_x, "pos")
14   <fftarray.Array (x: 2^2)> Size: 16 bytes
15   |dimension | space |    d     |   min    | middle   |   max    |  extent  |
16   +----------+-------+----------+----------+----------+----------+----------+
17   |    x     |  pos  |   0.67   |  -1.00   |   0.33   |   1.00   |   2.00   |
18   Values<jax.numpy>:
19   [0.  0.5 1.  1.5]
```

### 3.4.1 JAX Tracing

The tracing feature of JAX is often required to reach high computational performance when using JAX as the xp. Tracing extracts the computation graph of a Python function by executing it with placeholder values that retain only the shape and data type of arrays, not their actual values. This graph is then compiled for efficient execution, particularly on GPUs. It can also be modified to enable features like gradient computation. For tracing to work, JAX requires custom types (e.g. `Array` and `Dimension`) to mark which members are dynamic (replaced by placeholders) and which are static during computation.

FFTArray supports JAX tracing by providing the necessary implementations. Users must register `Array` and `Dimension` as JAX-compatible data structures by calling `fa.jax_regi⌋ ster_pytree_nodes()` before use.

By default, all members of the `Dimension` class are marked as static during tracing. This inserts all member values directly into the generated computation graph, enabling efficient reuse of compiled code (e.g., in loops). However, this prevents dynamic updates to grids during execution, as changes would require re-tracing the function. To enable dynamic grids (e.g., for moving domains) the creation functions of `Dimension` have the parameter `dynam⌋ ically_traced_coords`. Setting it to `True` makes $x_{\min}$, $f_{\min}$ and $\Delta x$ as well as all derived parameters except for $N$ (since JAX requires fixed shapes) dynamic at trace time. This allows to reuse the same function for different `Dimension`s but comes with a restriction in usability. Since most properties of `Dimension` are dynamic in this case, it cannot be checked at trace time whether one `Dimension` is equal to another. Therefore if two arrays each contain a `Di⌋ mension` with the same name but different tracers, they cannot be combined with each other as shown below:

```
1    import pytest
2    import fftarray as fa
3    import jax
4    fa.jax_register_pytree_nodes()
5    fa.set_default_xp(jax.numpy)
6
7    dim_x = fa.Dimension("x", 4, 0.5, 0., 0., dynamically_traced_coords=True)
8
9    @jax.jit
10   def my_fun(dim1: fa.Dimension) -> fa.Array:
11       arr1 = fa.coords_from_dim(dim1, "pos")
12       arr2 = fa.coords_from_dim(dim1, "pos")
13
14       # Works, because both arrays use the same dimension with the same tracers.
15       return arr1+arr2
16
17   my_fun(dim_x)
18
19   @jax.jit
20   def my_fun_not_dynamic(dim1: fa.Dimension, dim2: fa.Dimension) -> fa.Array:
21       arr1 = fa.coords_from_dim(dim1, "pos")
22       arr2 = fa.coords_from_dim(dim2, "pos")
23
```

```
24      # Addition requires all dimensions with the same name to be equal, this is
        ↪   explicitly checked before the operation.
25      # The check for equality fails with a `jax.errors.TracerBoolConversionError`
        ↪   because the coordinate grids' values of the `Dimension`s are only known at
        ↪   runtime.
26      # If `dynamically_traced_coords` above were set to False, the exact values of
        ↪   `dim1` and `dim2` were available at trace time and therefore this addition
        ↪   would succeed.
27      return arr1+arr2
28
29
30
31  with pytest.raises(jax.errors.TracerBoolConversionError):
32      my_fun_not_dynamic(dim_x, dim_x)
```

This can be solved by passing each `Dimension` instance exactly once into the jitted function. Note that when passing the same `Dimension` object as part of two different `FFTArray` objects, each `Dimension` instance gets its own distinct tracer. For example two `FFTArray` objects which contain a `Dimension` named `"x"` could not be combined inside a jitted function if they were passed in as parameters. Using `dynamically_traced_coords=True` requires very careful engineering of the code. Therefore, it defaults to `False` to cover the more common cases of static coordinate grids.

## 4 Examples

In this section, we demonstrate applications of FFTArray using various examples where gdFTs come into play. Section 4.1 demonstrates how to numerically compute a derivative. Section 4.2 describes how to use the split-step Fourier method to solve the Schrödinger equation. This method is then used in section 4.3 to simulate a matter-wave beam splitter using Bragg diffraction. Section 4.4 and Section 4.5 use a variation of Fourier split-step called imaginary time evolution to find the ground state of matter waves in a harmonic trap. Section 4.4 implements a single species wave function without self-interaction in a two-dimensional isotropic harmonic oscillator and evaluates the precision of the solution against the precise analytic solution. Section 4.5 extends that to two interacting Bose-Einstein condensates in a harmonic trap.

### 4.1 Derivative

The Fourier transform can be used to compute the n-th order derivative of a function $g(x)$ : $\mathbb{R} \to \mathbb{C}$ with:

$$\frac{\partial^n}{\partial x^n} g(x) = \widehat{\mathcal{F}} \{(2\pi i f)^n \mathcal{F} \{g(x)\}\}. \tag{44}$$

Note that directly discretizing this relation as shown in this chapter is only one way to numerically compute a derivative and roughly equivalent to a highest-order difference formula. If the signal showcases strong discontinuities including at the periodic boundaries of the sampled domain, other approaches like a lower order differencing formula can lead to better results. Such approaches can also be implemented with an FFT by a convolution with a different kernel [28, 29]. We showcase the implementation of eq. (44) with a modulated

Gaussian and both its analytic and numeric derivative:

$$g(x) = \cos(x) \, e^{\frac{-(x-1.25)^2}{25}}, \tag{45}$$

$$\frac{\partial}{\partial x} g(x) = \left( \frac{-2(x - 1.25)}{25} \cos(x) - \sin(x) \right) e^{\frac{-(x-1.25)^2}{25}}. \tag{46}$$

This test function and the x grid in the example code are both not symmetric around zero in order to show a general case where the the phase factors cannot be simplified. An important property of the test function is that it goes to zero on the edges of the sampled domain. If one extends the domain on both sides to get values on the boundaries even closer to zero, the precision of the derivative increases further in this case. The analytical function and its derivative are plotted alongside their numerical counterparts in fig. 3.

```python
import numpy as np
import fftarray as fa

# Test function and its derivative
g = lambda x: fa.cos(x)*fa.exp(-(x-1.25)**2/25.)
g_d1 = lambda x: ((-(2*(x-1.25))/25.)*fa.cos(x) -
    fa.sin(x))*fa.exp(-(x-1.25)**2/25.)

dim_x = fa.dim_from_constraints("x", # dimension name
    pos_min=-40., pos_max=50., d_pos=.5, # position space grid
    freq_middle=0., # frequency grid offset
    loose_params=["d_pos"], # The resulting d_pos in dim_x will be made smaller
        than the input d_pos such that N is a power of two.
)
x = fa.coords_from_dim(dim_x, "pos") # position space coordinate grid
f = fa.coords_from_dim(dim_x, "freq") # frequency space coordinate grid
sampled_fn = g(x) # sample the function in position space

# Compute the derivative
order = 1 # Order of the derivative
derivative_kernel = (2*np.pi*1.j*f)**order
g_d1_numeric = (sampled_fn.into_space("freq")*derivative_kernel).into_space("pos")

# Compute the expected result directly from the analytic derivative.
d1_analytic = g_d1(x)

# Compare the numeric and analytical result.
# In this example with these domains they are equal to at least eleven decimal
    digits.
np.testing.assert_array_almost_equal(g_d1_numeric.values("pos"),
    d1_analytic.values("pos"), decimal=11)
```

The `assert` in line 27 shows that this example is precise to up to 11 decimal digits. If the zero padding on the sides is chosen larger, the precision can also be higher. In this example the cancellation of the phase factors (reduced to `fftshift` and `ifftshift`) in section 2.3.4 happens automatically in line 20.

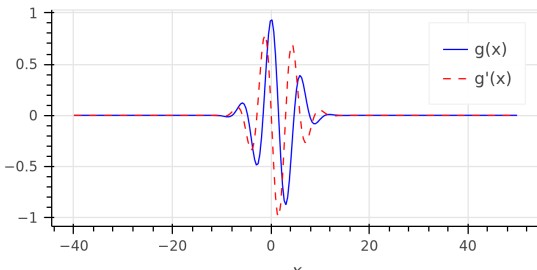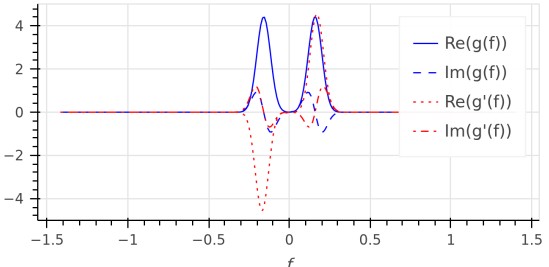

Figure 3: For illustration purposes a plot of the example function, eq. (45), and its first derivative, eq. (46), used to demonstrate differentiation using the gdFT. The left plot is in position space and the right plot in frequency space. Both are plotted in the exact domain and sample density used in the example code.

## 4.2 Solving the Schrödinger Equation

The Schrödinger equation is the central wave equation of quantum mechanics describing the time evolution of a single particle:

$$T_{\mathbf{r}} := \frac{-\hbar^2 \nabla_{\mathbf{r}}^2}{2m} \tag{47}$$

$$i\hbar \frac{\partial}{\partial t} \Psi(\mathbf{r}, t) = H(\mathbf{r}, t) \Psi(\mathbf{r}, t) \tag{48}$$

$$= (T_{\mathbf{r}} + V(\mathbf{r}, t)) \Psi(\mathbf{r}, t) \tag{49}$$

where $T_{\mathbf{r}}$ is the kinetic energy operator and $\Psi(\mathbf{r}, t) : (\mathbb{R}^n, \mathbb{R}) \to \mathbb{C}$ represents the particle's wave function in a (possibly) time-dependent potential $V(\mathbf{r}, t) : (\mathbb{R}^n, \mathbb{R}) \to \mathbb{R}$. $m$ denotes the mass of the particle and $\hbar$ is the reduced Planck constant.

The solution for the time propagation of the Schrödinger equation is the evolution operator $U$:

$$\Psi(\mathbf{r}, t+s) = U(t+s, t) \Psi(\mathbf{r}, t) \tag{50}$$

$$= \mathcal{T} \exp\left(-\frac{i}{\hbar} \int_t^{t+s} dt' s\, H(\mathbf{r}, t')\right) \Psi(\mathbf{r}, t) \tag{51}$$

$$\text{with } \mathcal{T}\left(A(t)B(t')\right) = \begin{cases} A(t)B(t'), & \text{if } t > t' \\ B(t')A(t), & \text{if } t < t' \end{cases} \tag{52}$$

Now we discretize the time evolution and only use very small time steps $\Delta t$ under the assumption that the Hamiltonian does not change too much over that time span. For a fixed time $t$ we get a time-independent $H(\mathbf{r}, t)$ under which we evolve for some time step $\Delta t$:

$$U(t + \Delta t, t) \Psi(\mathbf{r}, t) \approx \exp\left(-\frac{i}{\hbar} H(\mathbf{r}, t) \Delta t\right) \Psi(\mathbf{r}, t). \tag{53}$$

To achieve a longer time evolution each of these time steps is repeated multiple times to approximate the target dynamics.

For many problems the analytical evaluation of this solution is impractical. To solve it numerically, the wave function and potential can be approximated by sampling the wave function in position space at a high resolution. Evaluating the derivative contained in $H(\mathbf{r}, t)$ would then require a finite difference approximation. Exponentiating that finite difference approximation requires representing the resulting operator as a matrix with $\mathcal{O}(N^2)$ elements for the size $N$ of each dimension.

To avoid storing and multiplying matrices we can make use of eq. (44) and use a Fourier transform to turn the position derivative into a simple multiplication with $\mathbf{f}$:

$$\mathcal{F}\left(\nabla_{\mathbf{r}}^2\Psi(\mathbf{r})\right) = (2\pi\mathbf{f})^2\,\Psi(\mathbf{f}). \tag{54}$$

This process is called diagonalization and causes the matrix representation of the operator in our discrete basis to become diagonal and the exponential of a diagonal matrix is just the exponential of each of its diagonal element. However, the exponential also contains the potential operator. That operator is diagonal in position space and would become a non-diagonal derivative when transformed into frequency space. The evolution operator in eq. (53) can be split into an approximate product of three diagonal operators with a second order Trotter approximation, also called split-step or Strang-Splitting [1,2]:

$$\exp\left(-\frac{i}{\hbar}H(\mathbf{r},t)\Delta t\right)\Psi(\mathbf{r},t)$$
$$= \exp\left(-\frac{i}{\hbar}V(\mathbf{r},t)\frac{\Delta t}{2}\right)\exp\left(-\frac{i}{\hbar}T_{\mathbf{r}}\Delta t\right)\exp\left(-\frac{i}{\hbar}V(\mathbf{r},t)\frac{\Delta t}{2}\right)\Psi(\mathbf{r},t) + \mathcal{O}(\Delta t^3) \tag{55}$$

or alternatively

$$\exp\left(-\frac{i}{\hbar}H(\mathbf{r},t)\Delta t\right)\Psi(\mathbf{r},t)$$
$$= \exp\left(-\frac{i}{\hbar}T_{\mathbf{r}}\frac{\Delta t}{2}\right)\exp\left(-\frac{i}{\hbar}V(\mathbf{r},t)\Delta t\right)\exp\left(-\frac{i}{\hbar}T_{\mathbf{r}}\frac{\Delta t}{2}\right)\Psi(\mathbf{r},t) + \mathcal{O}(\Delta t^3). \tag{56}$$

The error analysis for this method has been carried out in [30–36]. The following will use eq. (55) since it is more efficient in section 4.5. With this approximation it is possible to make the kinetic energy operator diagonal in frequency space after a Fourier transform:

$$\mathcal{F}(T_{\mathbf{r}}) = \frac{\hbar^2}{2m}(2\pi\,\mathbf{f})^2, \tag{57}$$

$$\Rightarrow \mathcal{F}\left(\exp\left(-\frac{i}{\hbar}T_{\mathbf{r}}\frac{\Delta t}{2}\right)\right) = \exp\left(-\frac{i}{\hbar}\left(\frac{\hbar^2}{2m}(2\pi\mathbf{f})^2\right)\frac{\Delta t}{2}\right) \tag{58}$$

$$= \exp\left(-i\frac{\hbar}{2m}\frac{\Delta t}{2}(2\pi\mathbf{f})^2\right). \tag{59}$$

These split operators can be applied to $\Psi(\mathbf{r},t)$ by transforming it via the gdFT between position and frequency space in $\mathcal{O}(N\log N)$ time before applying each operator. Therefore a full time step can be implemented with $\mathcal{O}(N\log N)$ time complexity by transforming $\Psi(\mathbf{r},t)$ between the two spaces repeatedly:

$$\Psi_1(\mathbf{f}) = \Psi_0(\mathbf{f})\exp\left(-i\frac{\hbar}{2m}\frac{\Delta t}{2}(2\pi\mathbf{f})^2\right), \tag{60}$$

$$\Psi_2(\mathbf{r}) = \widehat{\mathcal{F}}(\Psi_1(\mathbf{f}))\exp\left(-i\frac{1}{\hbar}\Delta t\,V(\mathbf{r},t)\right), \tag{61}$$

$$\Psi_3(\mathbf{f}) = \mathcal{F}(\Psi_2(\mathbf{r}))\exp\left(-i\frac{\hbar}{2m}\frac{\Delta t}{2}(2\pi\mathbf{f})^2\right). \tag{62}$$

With FFTArray these formulas can be translated almost line by line into code to implement a single split-step Fourier time step of $\Delta t$:

```
from scipy.constants import hbar
import numpy as np
import fftarray as fa
```

```python
 4
 5   def split_step(psi0: fa.Array, *,
 6                  dt: float,
 7                  mass: float,
 8                  V: fa.Array,
 9                  ) -> fa.Array:
10       k_sq = 0.
11       for dim in psi0.dims:
12           # Using coords_from_arr ensures that attributes
13           # like eager and xp do match the ones of psi.
14           k_sq = k_sq + (2*np.pi*fa.coords_from_arr(psi0, dim.name, "freq"))**2
15
16       psi1 = psi0.into_space("freq") * fa.exp((-1.j * hbar/(2*mass) * dt/2) * k_sq)
17       psi2 = psi1.into_space("pos") * fa.exp((-1.j * hbar * dt) * V)
18       psi3 = psi2.into_space("freq") * fa.exp((-1.j * hbar/(2*mass) * dt/2) * k_sq)
19       return psi3
```

The for-loop to compute `k_sq` and the automatic vectorization of arithmetic expressions enable this whole snippet to automatically support multiple dimensions and lazy evaluation (see section 3.3) automatically skips unnecessary phase factors. If `psi0` has `factors_applied` `=False`, the whole `split_step` routine never applies a single set of scale and phase factors, because each operator application is only a multiplication. Therefore, calling `split_step` multiple times in a loop does not have any per-step overhead while still supporting arbitrarily shifted coordinate grids. Every space change is just the call to `fft` or `ifft`, respectively. The `into_space` function allows the user to pass in `psi0` in any space and with any value for `factors_applied`. Any necessary transformations are done automatically. This and all following examples implement their calculations in SI units.

The split-step method can be modified to find the lowest energy eigenstate of an arbitrary potential. This so-called imaginary time evolution is achieved by replacing the time step $\Delta t$ with an imaginary time step $\Delta t \mapsto -i\Delta t$ such that the time evolution operator becomes $\exp\left(-\frac{1}{\hbar}H(\mathbf{r}, t)\Delta t\right)$. This causes each time step to dampen eigenstates with higher eigenenergies faster than the ones with lower energies. Since the whole wave function is dampened with every step, it would quickly become too small to be representable with the used floating point numbers. To prevent that it has to be renormalized after each time step.

We collected helper functions and definitions which are specific to quantum mechanical matter wave problems in a separate package called `matterwave` [37]. It contains an implementation of the split-step algorithm, often used constants and helper functions for normalizing wave functions and calculating expectation values and kinetic energies.

## 4.3 Bragg Diffraction of Matter Waves

In this example we solve the Schrödinger equation with the split-step method from section 4.2 to simulate matter wave diffraction in a Bragg grating made of light. Bragg diffraction is one of the central atom optical operations in atom interferometry [38, 39] to transfer several photon recoils of momentum without changing the internal state of the atoms [40–43]. This can be used to create a superposition of momentum states and thus is also referred to as a beam splitter. This example implements a semi-classical model of Bragg diffraction. The atom is described by a quantum-mechanical wave function. Due to its high enough intensity the light field is described classically since there are always enough photons [38, 44, 45]. Two counter-propagating laser beams form a lattice which causes elastic scattering of matter waves. This process does not change the internal state of the atom and the whole dynamics are described

with the following Hamiltonian [16]:

$$H(x,t) = -\frac{-\hbar^2}{2m}\frac{\partial^2}{\partial x^2} + 2\hbar\Omega(t)\cos^2(k_L x - 2\omega_r t) \tag{63}$$

where $\Omega(t)$ is the time-dependent effective Rabi frequency. $\Omega(t)$ is determined by the laser properties and is proportional to the intensity of the laser fields. The mass $m$ in this example is for $^{87}$Rb and $k_L$ is the wavenumber of the utilized atomic transition which is in this case the D2 line with a wavelength of $780\,\mathrm{nm}$ [46]. The single-photon recoil frequency of this wavelength is then $\omega_r = \hbar k_L^2/2m$. The initial state is defined to be a Gaussian with a momentum width of $0.01\hbar k_L$. The code reuses the `split_step` function of section 4.2 and the actual potential is simply the potential part of the Hamiltonian in eq. (63). Depending on the passed in ramp, this code simulates all Bragg regimes from Deep-Bragg to Raman-Nath as shown in [16].

```python
import numpy as np
from scipy.constants import hbar

# Rb87 mass in kg
mass_rb87: float = 86.909 * 1.66053906660e-27
# Rb87 D2 transition wavelength in m
lambda_L: float = 780 * 1e-9
# Bragg beam wave vector
k_L: float = 2 * np.pi / lambda_L
hbark: float = hbar * k_L
# Single-Photon recoil frequency
w_r = hbar * k_L**2 / (2 * mass_rb87)

def simulate_bragg(t_arr, dt: float, rabi_frequency, ramp_arr, xp=np,
    dtype=np.float64):
    # Returns the wave function after applying the Bragg beam potential to a
    #   Gaussian input state with 0.01 hbark initial momentum width. The beam is
    #   assumed to be spatially homogeneous.
    # t_arr: NumPy array containing the t of each time step.
    # dt: Size of a time step
    # rabi_frequency: Sets the magnitude of Ω(t), determined by the concretely
    #   used atom transition and laser detuning and intensity.
    # ramp_arr: Scaling factor for Ω(t) for each time step.
    # Ω(t) = rabi_frequency*ramp_arr

    # Dimension for full sequence based on expected matter wave size and expansion
    #   speed
    dim_x: fa.Dimension = fa.dim_from_constraints("x",
        pos_extent = 2e-3,
        pos_middle = 0,
        freq_middle = 0.,
        freq_extent = 32. * k_L/(2*np.pi),
        loose_params = ["freq_extent"]
    )
    # Initialize array with position coordinates.
    x: fa.Array = fa.coords_from_dim(dim_x, "pos", xp=xp, dtype=dtype)

    # Initialize harmonic oscillator ground state
    sigma_p=0.01*hbark
    psi: fa.Array = (2 * sigma_p**2 / (np.pi*hbar**2))**(1./4.) * \
        fa.exp(-(sigma_p**2 / hbar**2) * x**2)
    # Numerically normalize so that the norm is `1.` even though the tails of the
    #   Gaussian are cut off.
    psi *= fa.sqrt(1./fa.integrate(fa.abs(psi)**2))

    # For each time step, compute the potential and evolve the wave function in it
```

```
41      for t, ramp in zip(t_arr, ramp_arr):
42          V = rabi_frequency * ramp * 2. * hbar * fa.cos(
43              k_L * x - 2. * w_r * t
44          )**2
45          psi: fa.Array = split_step(
46              psi,
47              dt=dt,
48              mass=mass_rb87,
49              V=V,
50          )
51
52      return psi
```

Depending on the specific scientific scenario, this implementation of Bragg diffraction might need different inputs and control parameters. Please note that different array libraries can require modifications to this code for peak performance, e.g. the JAX library requires to replace the above for loop with its `jax.lax.scan` function, see also section 5.

### 4.3.1  Raman-Nath Regime

The Raman-Nath regime is characterized by a very short and bright pulse of a spatially symmetric beam splitter with a duration $\tau \ll \frac{1}{\sqrt{2\Omega\omega_r}}$. In this regime an analytical solution is available [38, 44]:

$$|g_n(t)|^2 = J_n^2(\Omega t) \tag{64}$$

where $g_n(t)$ is the amplitude of the $n$-th momentum state $|2n\hbar k\rangle$ and $J_n$ the Bessel functions of first kind. Following [16], their demonstration scenario for this case with $\Omega = 50\omega_r$ and $\tau = 1\,\mu s$ with a rectangular intensity profile in time can be implemented with the code below.

```
1  rabi_frequency = 50*w_r
2  n_steps = 200
3  t_arr, dt = np.linspace(0., 1e-6, n_steps, retstep=True, endpoint=False)
4  ramp_arr = np.full(n_steps, 1.)
5
6  psi = simulate_bragg(t_arr, dt, rabi_frequency, ramp_arr)
```

The results of this code are visualized in fig. 4 and show very good agreement with the analytical solution.

### 4.3.2  Bragg Regime

The laser intensity in a two-(momentum)-level beam splitter typically has a Gaussian temporal profile to ensure broad velocity selectivity such that most atoms participate in the Rabi oscillation between the two momentum states. The below code snippet implements a Gaussian temporal profile with $\sigma = 25\,\text{ms}$, optimized to simulate a two-level single Bragg diffraction beam splitter in 401 steps. Special care was taken to sample the temporal profile symmetrically around $t = 0$ while explicitly sampling the peak intensity.

```
1  # Use an odd number of steps to symmetrically sample the Gaussian
2  # and hit its peak at t=0 with a sample.
3  # Note that this snippet does not start at t=0 like above, but is symmetric around
   ↪  t=0.
4  n_steps = 401
5  # Rabi frequency. This specific value was found as a binary search to
6  # optimize a 50/50 split of the two momentum classes for this specific beam
```

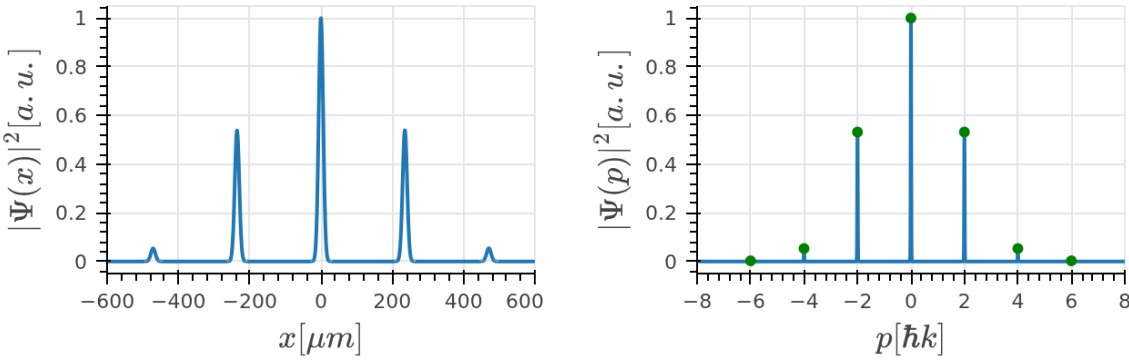

Figure 4: Probability density in position and momentum space after a Raman-Nath pulse with a rectangular temporal profile and $\Omega = 50\omega_r, \tau = 1\,\mu s$ followed by 20 ms of free propagation like in [16]. The green dots mark the analytical solution from eq. (64).

```
7   # splitter duration and pulse form.
8   rabi_frequency = 25144.285917282104 # Hz
9   sigma_bs = 25e-6 # temporal pulse width (s)
10  # The Gaussian is sampled  from -4*sigma_bs to 4*sigma_bs
11  sampling_range_mult = 4.
12  t_arr, dt = np.linspace(
13      start=-sampling_range_mult*sigma_bs,
14      stop=sampling_range_mult*sigma_bs,
15      num=n_steps,
16      retstep=True,
17  )
18  # Gaussian density function
19  gauss = lambda t, sigma: np.exp(-0.5 * (t / sigma)**2)
20  # Remove the value of the Gauss at the beginning of the pulse so that
21  # the intensity starts and ends at zero.
22  gauss_offset = gauss(t = t_arr[0], sigma = sigma_bs)
23  ramp_arr = gauss(t = t_arr, sigma = sigma_bs) - gauss_offset
24
25  psi = simulate_bragg(t_arr, dt, rabi_frequency, ramp_arr)
```

The results of this beam splitter are visualized in fig. 5. It shows that the initial atom wave function with an average momentum of $0\hbar k$ was split cleanly into a superposition of two momentum classes of 0 and $2\hbar k$. The free propagation made this momentum split also visible in position space.

## 4.4 Finding the Ground State of the Two-Dimensional Isotropic Quantum Harmonic Oscillator

The quantum harmonic oscillator is a central model system of quantum mechanics because it can be used to approximate many other systems. It is one of the few systems for which an exact, analytical solution for its eigenstates and eigenvectors is known. This makes it a very good opportunity to compare the numerical precision of a simple solver based on FFTArray with its exact solution. The isotropic quantum harmonic oscillator in $n$ dimensions is defined as:

$$H = \frac{-\hbar^2 \nabla_{\mathbf{r}}^2}{2m} + \frac{1}{2}m\omega^2 \mathbf{r}^2, \quad \mathbf{r} \in \mathbb{R}^n \qquad (65)$$

In this case, the angular frequency $\omega$ of the oscillator is the same in all directions.

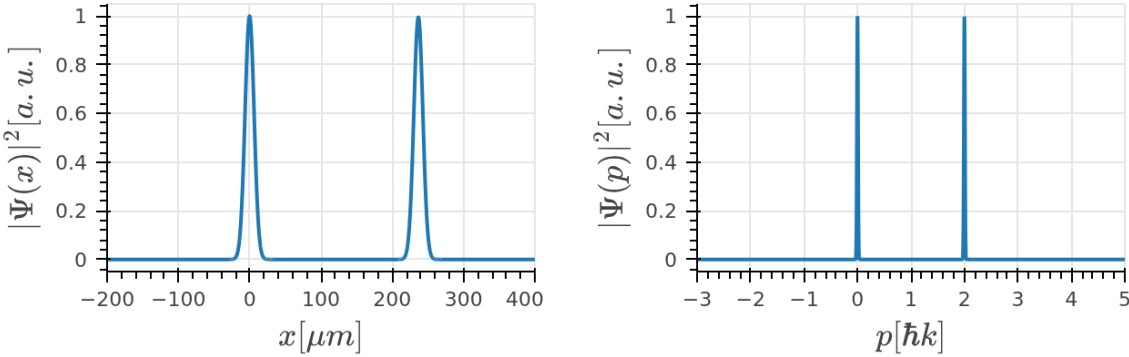

Figure 5: Probability density in position and momentum space of a Gaussian wave function with initial width of $\Delta p = 0.01\hbar k$ after Bragg beam splitter with a $\sigma = 25\,\text{ms}$ Gaussian temporal profile followed by $20\,\text{ms}$ of free propagation. The 50/50 split between the two momentum classes $|g, 0\rangle$ and $|g, 2\hbar k_L\rangle$ works very well and these ideal parameters of long smooth pulses and very sharp peaks in momentum space do not yet show visible velocity selectivity.

The solution for its ground state energy is

$$E_0 = \hbar\omega\frac{n}{2}. \tag{66}$$

The following is a direct implementation of the imaginary time evolution described in section 4.2 for the isotropic quantum harmonic oscillator in two dimensions:

```python
omega = 0.5*2.*np.pi

dim_x = fa.dim_from_constraints("x",
    pos_min=-100e-6,
    pos_max=100e-6,
    freq_middle=0.,
    n=2048,
)
y_dim = fa.dim_from_constraints("y",
    pos_min=-100e-6,
    pos_max=100e-6,
    freq_middle=0.,
    n=2048,
)

V: fa.Array = 0. # type: ignore
for dim in [dim_x, y_dim]:
    V = V + 0.5 * mass_rb87 * omega**2. * fa.coords_from_dim(dim, "pos")**2

k_sq = 0.
for dim in [dim_x, y_dim]:
    k_sq = k_sq + (2*np.pi*fa.coords_from_dim(dim, "freq"))**2

# Initialize psi as a constant function with value 1.
psi = fa.full(dim_x, "pos", 1.) * fa.full(y_dim, "pos", 1.)
for _ in range(n_steps):

    psi = psi.into_space("pos") * fa.exp((-0.5 / hbar * dt) * V)
    psi = psi.into_space("freq") * fa.exp((-1. * dt * hbar / (2*mass_rb87)) *
    ↪ k_sq)
    psi = psi.into_space("pos") * fa.exp((-0.5 / hbar * dt) * V)

```

```
32        state_norm = fa.integrate(fa.abs(psi)**2)
33        psi = psi * fa.sqrt(1./state_norm)
```

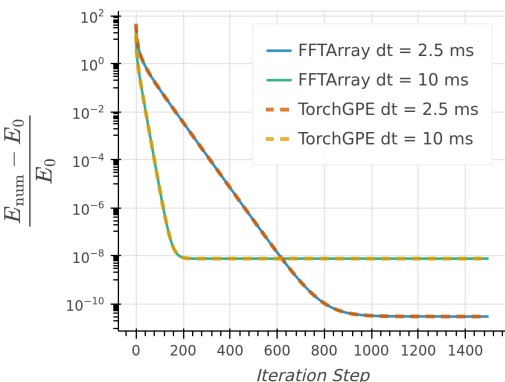

Figure 6: Absolute value of relative energy difference between the analytical and the numerical ground state of a 2D isotropic quantum harmonic oscillator for FFTArray and TorchGPE in float64. For both implementations, smaller time steps converge slower but are also able to more closely approach the analytic solution. With a time step of $dt = 2.5$ ms both implementations reach the ground state energy with a relative error better than $10^{-9}$.

The energy of a wave function is the sum of its potential and kinetic energy:

$$E_{\text{tot}} = E_{\text{kin}} + E_{\text{pot}}, \tag{67}$$

$$E_{\text{kin}} = \frac{\hbar^2}{2m} \int d^n\mathbf{f} \, |\Psi(\mathbf{f})|^2 (2\pi\mathbf{f})^2, \tag{68}$$

$$E_{\text{pot}} = \frac{\hbar^2}{2m} \int d^n\mathbf{r} \, |\Psi(\mathbf{r})|^2 V(\mathbf{r}). \tag{69}$$

As the metric for how well the found solution approximates the analytic solution we use the relative difference in energies between the numerical and analytical solution:

$$E_{\text{diff}} = \frac{E_{\text{num}} - E_0}{E_0} \tag{70}$$

with $E_{\text{num}}$ being the total energy of the numeric solution.

The resulting energies as a function of the number of time steps are shown in fig. 6. Smaller time steps converge slower but are able to reach the analytical solution more precisely. For reference we also added the results of an implementation with TorchGPE [11].

### 4.4.1 Single Precision Simulation (float32)

The results in fig. 6 are computed with double precision (float64) numbers. However, there are only few use cases which require such high precision outside of scientific computing. Therefore, many GPUs feature much higher single precision than double precision compute or even just single precision compute. Examples for this are most current consumer GPUs like the NVIDIA AD102 (RTX 4090) which typically have 64 times more float32 compute than float64 compute [47].

Each algorithm and scenario potentially require a different numerical precision. As shown in fig. 7a, the precision of the result is reduced by about 4 orders of magnitude for $dt = 2.5$ ms

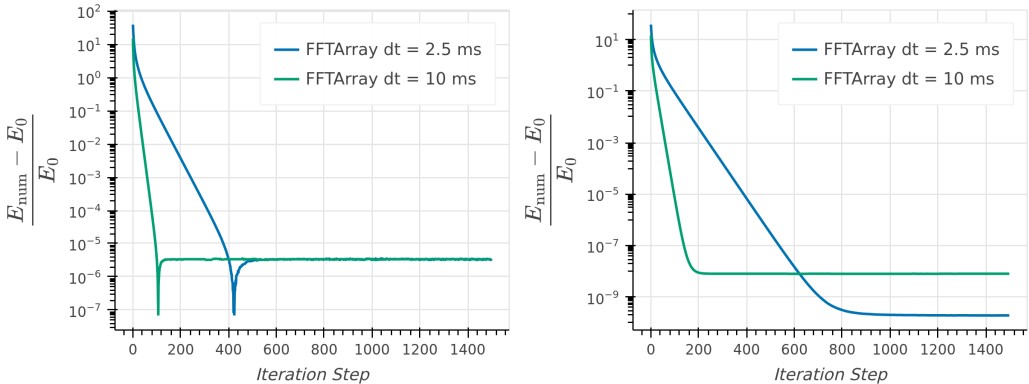

(a) Imaginary time evolution and energy evaluation in float32.

(b) Imaginary time evolution in float32 and energy evaluation in float64.

Figure 7: Relative energy difference between the analytical and the numerical ground state of a 2D isotropic quantum mechanic oscillator in single precision (float32) with FFTArray. On the left, the computation of the energy of the state is also done in float32 while on the right, only the imaginary time evolution is done in float32 while the energies are evaluated in float64. This hybrid approach shows an improvement of three orders of magnitude.

when doing the whole computation and evaluation in float32. The main limit in this case is not the actual imaginary time evolution but just the evaluation of the energy of the solution. When keeping the imaginary time evolution at float32 but computing the energy with float64 numbers, the precision is only reduced by about one order of magnitude compared to the float64 result. Implementing such hybrid algorithms is extremely easy with FFTArray because it only requires changing the data type of the `psi` array before passing it to the evaluation function.

## 4.5 Finding a Two-Species Ground State in a Harmonic Trap

In this example, we perform an imaginary time evolution to find the ground state of two interacting atomic species within harmonic potentials. In particular, we describe a system of two magnetically trapped Bose Einstein condensates (BECs) of $^{87}$Rb and $^{41}$K, relevant in fundamental physics experiments [48,49]. The traps can be described with three-dimensional anisotropic harmonic potentials with different trap frequencies for each of the two species. Typically, these form a cigar shape with two large frequencies in two dimensions and one small frequency in the third dimension.

We describe this quantum mechanical system via a coupled Gross-Pitaevskii equation (GPE). The GPE of a single species is a non-linear Schrödinger equation with a term that describes the self-interaction of the atoms. It is quantified by the scattering amplitudes $g_{Rb}$ and $g_K$, respectively. The coupled GPE additionally features an additional interaction between the two different species with the scattering amplitude $g_{Rb,K}$. The respective time-independent Hamiltonians for these wavefunctions are [15,50]:

$$H_{Rb}(\mathbf{r},t) = -\frac{\hbar^2}{2m_{Rb}}\nabla_{\mathbf{r}}^2 + V_{Rb}^{ext}(\mathbf{r}) + N_{Rb}\,g_{Rb}\,|\Psi_{Rb}(\mathbf{r})|^2 + N_K\,g_{Rb,K}\,|\Psi_K(\mathbf{r})|^2, \tag{71}$$

$$H_K(\mathbf{r},t) = -\frac{\hbar^2}{2m_K}\nabla_{\mathbf{r}}^2 + V_K^{ext}(\mathbf{r}) + N_K\,g_K\,|\Psi_K(\mathbf{r})|^2 + N_{Rb}\,g_{Rb,K}\,|\Psi_{Rb}(\mathbf{r})|^2, \tag{72}$$

with the respective number of atoms in the BEC $N_{Rb,K}$, the atom mass $m_{Rb,K}$ and a 3-dimensional anisotropic trapping potential $V_{Rb,K}^{ext}$ like in section 4.4 with a frequency relation

$$\omega_{\mathrm{K}}^i = (m_{\mathrm{Rb}}/m_{\mathrm{K}})^{1/2}\, \omega_{\mathrm{Rb}}^i.$$

Performing the imaginary time evolution for two coupled wave functions instead of a single one can be straightforwardly implemented by adapting 4.2. The total potential $V(\mathbf{r}, t)$ can be written directly as a sum of the external potential, the self-interaction and the interaction with the other species. This is possible as FFTArray offers full control over each time step while both, wave functions and operators, are free-standing `Array` objects which can be combined arbitrarily. The following shows a single imaginary time step of the above system:

```python
def imaginary_time_step_dual_species(
    psi_rb87: fa.Array,
    psi_k41: fa.Array,
    rb_potential: fa.Array,
    k_potential: fa.Array,
    dt: float,
) -> Tuple[fa.Array, fa.Array]:
    """
    Perform a single imaginary time step for the dual species GPE.
    """

    ## Calculate the potential energy operators (used for split-step and plots)
    psi_rb87 = psi_rb87.into_space("pos")
    psi_k41 = psi_k41.into_space("pos")

    psi_pos_sq_rb87 = fa.abs(psi_rb87)**2
    psi_pos_sq_k41 = fa.abs(psi_k41)**2

    self_interaction_rb87 = num_atoms_rb87 * coupling_rb87 * psi_pos_sq_rb87
    interaction_rb87_k41 = num_atoms_k41 * coupling_rb87_k41 * psi_pos_sq_k41
    V_rb87 = self_interaction_rb87 + interaction_rb87_k41 + rb_potential

    self_interaction_k41 = num_atoms_k41 * coupling_k41 * psi_pos_sq_k41
    interaction_k41_rb87 = num_atoms_rb87 * coupling_rb87_k41 * psi_pos_sq_rb87
    V_k41 = self_interaction_k41 + interaction_k41_rb87 + k_potential

    ## Imaginary time split step application

    psi_rb87 = split_step_imaginary_time(
        psi=psi_rb87,
        V=V_rb87,
        dt=dt,
        mass=m_rb87,
    )
    psi_k41 = split_step_imaginary_time(
        psi=psi_k41,
        V=V_k41,
        dt=dt,
        mass=m_k41,
    )

    return psi_rb87, psi_k41
```

```python
import jax.numpy as jnp
from functools import reduce

def split_step_imaginary_time(
    psi: fa.Array,
    V: fa.Array,
    dt: float,
    mass: float,
) -> fa.Array:
```

```
10      """Perform an imaginary time split-step of second order in VPV
        ↪ configuration."""

12      # Calculate half step imaginary time potential propagator
13      V_prop = fa.exp((-0.5*dt / hbar) * V)
14      # Calculate full step imaginary time kinetic propagator (k_sq = kx^2 + ky^2 +
        ↪ kz^2)
15      k_sq = reduce(lambda a,b: a+b, [
16          (2*np.pi * fa.coords_from_dim(dim, "freq", xp=jnp, dtype=jnp.float64))**2
17          for dim in psi.dims
18      ])
19      T_prop = fa.exp(-dt * hbar * k_sq / (2*mass))

21      # Apply half potential propagator
22      psi = V_prop * psi.into_space("pos")

24      # Apply full kinetic propagator
25      psi = T_prop * psi.into_space("freq")

27      # Apply half potential propagator
28      psi = V_prop * psi.into_space("pos")

30      # Normalize after step
31      state_norm = fa.integrate(fa.abs(psi)**2)
32      psi = psi / fa.sqrt(state_norm)

34      return psi
```

For the full code including the initialization of all functions and constants including energy tracking, we refer to the corresponding example in the repository [51]. Figure 8 shows the final probability densities for both species. They qualitatively match the results shown in [15] from where we adopted the system parameters.

Note that in this case it is beneficial to split the operator with two times the potential as in eq. (55). The potential has to be computed before applying the first of the split operators, in order to not degrade into an effectively first order propagation. With the above split (eq. (55)) of the evolution operator, the wave functions are already in position space before and after each time step which allows to directly compute the self-interaction from it. With the other split as in eq. (56), they would be in frequency space before a time step and the potential calculation would require the execution of two additional IFFTs per time step.

## 5 Computational Performance Evaluation

In the following, we evaluate the computational performance of FFTArray on one of our previous examples. We compare the array libraries NumPy and JAX as well as the execution on different hardware. Due to the breadth of possible implementations on top of FFTArray, any performance evaluation can only look at a small subset of possible scenarios. This section uses as an example the imaginary time evolution from section 4.4.

This section evaluates whether FFTArray maintains performance comparable to directly using the underlying array library's FFT methods. The comparison focuses on large domains and a split-step algorithm with many time steps since those are for us the most performance-limited scenarios. In order to focus on those, we exclude startup and initialization costs from the time measurement as much as possible. In order to extract the run-time per step and remove any remaining startup costs, we perform a linear fit of the results for three different numbers of time steps and take the fitted slope as our result in computational time per time

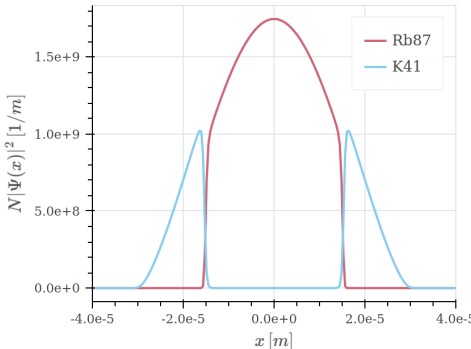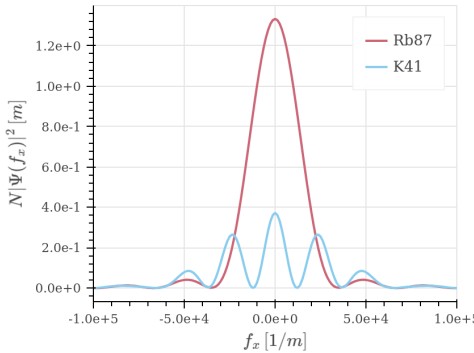

Figure 8: The computed ground state probability densities of the $^{87}$Rb and $^{41}$K BECs normalized to the number of atoms per species, shown in both position and frequency space. The repelling force between the two species splits up $^{41}$K along the weak trapping axis $x$. The plotted position and frequency space regions are zoomed in with respect to the actual domain sizes.

step. For more details on all these reduction steps see appendix A.2 and appendix A.3.

We compare three different implementations of the same imaginary time evolution. They all execute the exact same algorithm but organize the computations differently. The first one is the code from section 4.4 which will be labeled as "FFTArray Direct".

The potential and time step `dt` are the same for each step in this scenario. We can thus compute the propagators once before the loop and then reuse them at each step. This insight yields the second implementation which is called "FFTArray Precomputed":

```python
# <Same intialization as in the original example.>

V = get_V(psi)
k_sq = 0.
for dim in psi.dims:
    # Using coords_from_arr ensures that attributes
    # like eager and xp do match the ones of psi.
    k_sq = k_sq + (2*np.pi*fa.coords_from_arr(psi, dim.name, "freq"))**2

T_prop = fa.exp((-1. * dt * hbar / (2*mass)) * k_sq)
V_prop = fa.exp((-0.5 / hbar * dt) * V)

for _ in range(n_steps):
    psi = psi.into_space("pos") * V_prop
    psi = psi.into_space("freq") * T_prop
    psi = psi.into_space("pos") * V_prop

    state_norm = fa.integrate(fa.abs(psi)**2)
    psi = psi * fa.sqrt(1./state_norm)
```

In order to measure how much overhead FFTArray causes inside the loop, we create a a third benchmark variant which does not use any code of FFTArray inside the loop and is called "Raw FFT":

```python
# <Same intialization as in the above example.>

T_prop = fa.exp((-1. * dt * hbar / (2*mass)) * k_sq)
V_prop = fa.exp((-0.5 / hbar * dt) * V)

```

```
6   T_prop_arr = T_prop.values("freq")
7   V_prop_arr = V_prop.values("pos")
8   # Need the raw inner values, which are not accessible via a public API
9   # Using these inner values allows us to use the whole infrastructure and phase
    ↪  setup of FFTArray in this example.
10  # In the inner loop below FFTArray would not need to apply any phase and scale
    ↪  factors because they would cancel out.
11  psi = psi.into_space("pos").into_factors_applied(False)._values
12
13  for _ in range(n_steps):
14      psi *= V_prop_arr
15      psi = xp.fft.fftn(psi)
16      psi *= T_prop_arr
17      psi = xp.fft.ifftn(psi)
18      psi *= V_prop_arr
19
20      state_norm = xp.sum(xp.abs(psi)**2)*vol_elem
21      psi *= xp.sqrt(1./state_norm)
22
23  # Repack the raw values correctly.
24  # Again there is no public API for that.
25  psi = fa.array(values, [x_dim, y_dim], "pos")
26  psi._factors_applied = (False,)*len(dims)
```

This manually eliminates FFTArray completely from the inner loop in order to test whether the book-keeping of FFTArray creates measurable overhead when ensuring dimensions line up and phase factors are applied if necessary. Doing something like this in practice also abandons the advantages FFTArray offers, which is why this code snippet needs to access private implementation details of FFTArray.

These three implementations are measured with two different array libraries. NumPy is the base reference and only runs on CPUs. For GPU support and potentially less overhead on CPUs, we use the JAX library. They recommend using special structured control flow primitives [52] in order to achieve best performance. Therefore, the `for` loop is replaced with `jax.lax.s`‿`can` for these measurements. This ensures that JAX can optimize the whole loop as one unit and operations are potentially fused which can increase performance and reduce the per-step overhead.

In order to give a broad overview of the implementations on different hardware, we chose one server and one desktop CPU (AMD Epyc 7543, AMD Ryzen 7950X3D) and GPU (NVIDIA A100, NVIDIA RTX 4090) respectively, for more details see appendix A.1.

These measurements were done at a resolution of 4096 by 4096 samples such that the values reflect the speed in the limit of large wave functions as well as possible. With this number of samples, the complex-valued wave function has a size of 128 MiB (float32) and 256 MiB (float64), respectively, which does not fit in any of the caches of the tested processors. Other domain sizes can be estimated roughly by linearly extrapolating, for details to the scaling as a function of domain size and shape, see appendix A.4.

The results in fig. 9 show the extremely large performance difference between CPUs and GPUs for this example. The JAX implementation is about two orders of magnitude faster in most cases on the two GPUs compared to the two CPUs. This shows that it is worthwhile to use the consumer GPUs over any kind of CPUs even if the computations have to happen in float64.

The tracing and batching of the `scan`-function of JAX is able to achieve almost the same speed in all three implementations on both CPUs and GPUs. It therefore shows that FFTArray does not add any measurable overhead compared to using the underlying array library directly for large arrays and time steps. This means, when using JAX, users can even take advantage of the comfort of directly writing down the formulas without a significant hit to performance

on any of the tested platforms.

The NumPy implementation only runs on the CPUs and is slower than JAX on the same hardware. "FFTArray Direct" is significantly slower than the other two implementations. This is expected since NumPy cannot fuse multiple operations together which causes a much higher memory overhead for recomputing the propagators at each time step. However, when precomputing the propagators in "FFTArray Precomputed" the overhead for managing dimensions and phase factors becomes small in the tested scenario and this implementation achieves comparable speeds to the "Raw FFT" variant. Therefore, our goal of not introducing overhead for our key use-cases is also achieved with NumPy.

The A100 is faster than the RTX 4090 in float64 while it is the other way around in float32. This difference is less pronounced than their huge differences in peak compute performance would suggest. Performance is most likely bound more by memory accesses than raw compute performance in many parts of the algorithm.

In order to give a comparison to an existing state-of-the-art solution, measurement results of TorchGPE are listed at the bottom of the chart for the same hardware. The resolution in position space is the same as in the FFTArray implementation. However, it uses twice as many samples in frequency space to reduce boundary effects [53]. As shown in fig. 6 this does not yield a significantly better precision in this scenario. Since this feature cannot be deactivated at the time of writing, it is part of the overhead of TorchGPE in this scenario. Even if its performance was a factor of two better, our solution is still multiple times faster in the measured scenario.

# 6 Conclusion and Outlook

FFTArray introduces a framework for implementing discretized Fourier transforms on arbitrarily shifted coordinate grids. It enables researchers to translate Fourier integral formulas directly into code and easily scales from single to multiple dimensions via named dimensions. Numerical details like choosing valid coordinate grids and implementing all necessary phase and scale factors to obtain a discretized Fourier transform are handled automatically without performance overhead for large simulations. By being built upon the Python Array API standard, FFTArray enables high performance on GPUs that reduces simulation run times significantly and makes large-scale 3D simulations computationally feasible. This was already utilized successfully with in-development versions of FFTArray for multiple scientific publications [54–57].

FFTArray allows researchers to focus on the core scientific challenges rather than the intricacies of Fourier transform implementations, therefore enabling the rapid prototyping of complex models.

Many of the library's central ideas like its constraint solver, the encapsulation of coordinates per dimension and giving the user explicit but automated control over phase and scale factor applications are language-independent and may be adapted to other programming languages. We hope that FFTArray's modular architecture and design will empower researchers to tackle Fourier-related challenges more effectively. Furthermore, we invite the scientific community to expand upon our existing implementations, such as our matter wave simulation package [37], as well as to contribute to this package and build new solvers on top it.

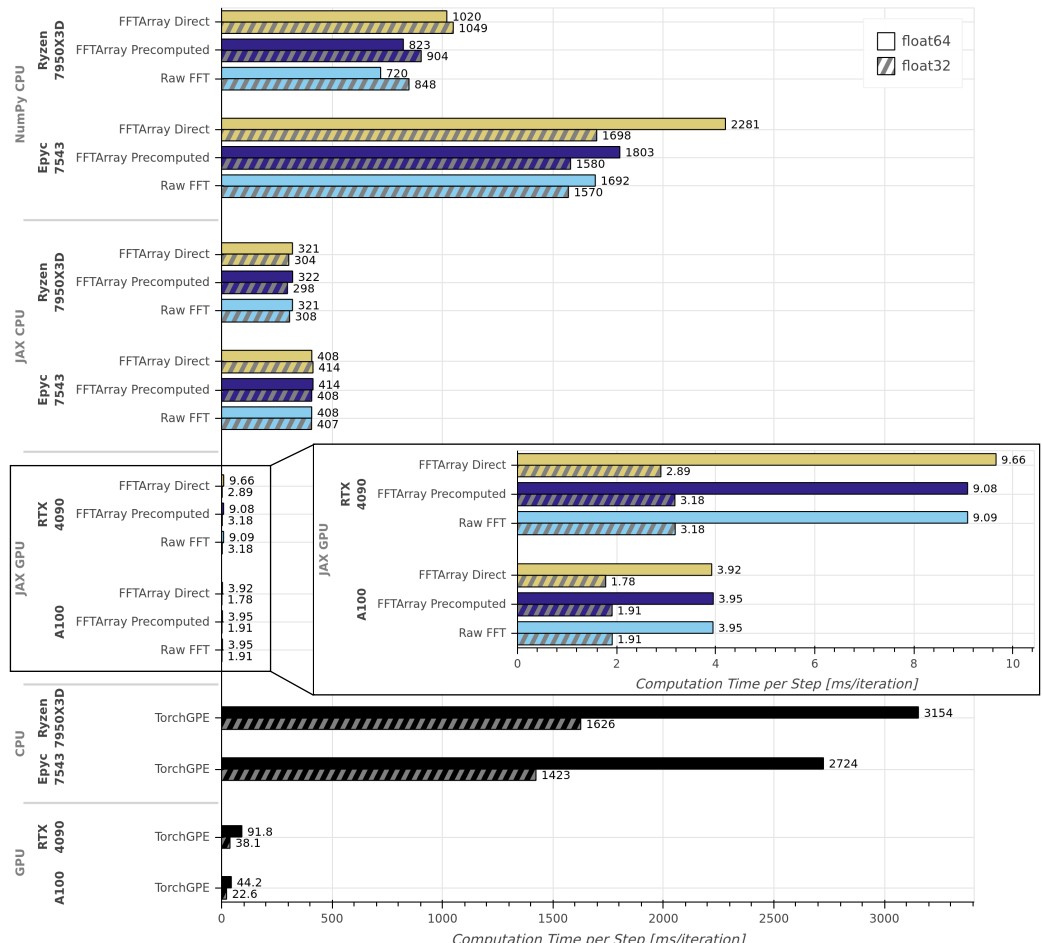

Figure 9: Computation time per time step on the problem of finding the ground state of an isotropic 2D quantum harmonic oscillator. This compares different loop implementations and hardware at a resolution of 4096 by 4096 samples. The similar performance between "Raw FFT" and "FFTArray Precomputed" demonstrates that FFTArray introduces negligible overhead. GPUs are about two orders of magnitude faster than CPUs and should therefore be preferred in most cases. The comparison with TorchGPE shows that the performance of this implementation is better than other state-of-the-art solutions in this scenario.

**Code Availability**  The code of FFTArray is openly available at https://github.com/QSTheory/fftarray and our matter wave specific library is openly available at https://github.com/QSTheory/matterwave, both under Apache-2.0 license.

# Acknowledgements

We thank our science group for feedback on earlier versions of this package and helping us charting out the variety of use cases for this package. S.J.S. thanks Florian Fitzek for his introduction, guidance and Fortran codes for the simulation of matter wave interferometers. We thank Eric Charron for his feedback on an intermediate version of the library. S.J.S. thanks Alexander Hahn for discussions, guidance about Trotter product formulas and split-step algorithms and very helpful feedback to a draft of this paper. G.M. thanks Annie Pichery for discussions about the two species ground state example.

**Funding information**   This work was funded by the Deutsche Forschungsgemeinschaft (German Research Foundation) under Germany's Excellence Strategy (EXC-2123 QuantumFrontiers Grants No. 390837967), through CRC 1227 (DQ-mat) within Projects No. A05 and the German Space Agency (DLR) with funds provided by the German Federal Ministry for Economic Affairs and Climate Action (BMWK) (German Federal Ministry of Education and Research (BMBF)) due to an enactment of the German Bundestag under Grants No. 50WM2245A (CAL-II), No. 50WM2263A (CARIOQA-GE) and No. 50WM2253A (AI-Quadrat). NG acknowledges funding by the AGAPES project - grant No 530096754 within the ANR-DFG 2023 Programme. J.-N. K.-S., G.M., C.S. and S.J.S. acknowledge support from QuantumFrontiers through the QuantumFrontiers Entrepreneur Excellence Programme (QuEEP). J.-N. K.-S., G.M., S.J.S. and N.G. acknowledge funding from the EU project CARIOQA-PMP (101081775).

# A  Computation Speed as a Function of Time Steps and Function Samples

This section describes in more detail the methods, hardware and software used to create the speed measurements in section 5. In order to generate the numbers shown in fig. 9, a few assumptions about the distribution of our measurement data were made, like the linearity of simulation time as a function of time steps. These are also described and justified in this section.

The measured simulation is an imaginary time evolution to find the ground state of an n-dimensional harmonic oscillator as described in sections 4.2 and 4.4. This means that among others, there are the following numerical parameters:

- The number of iterations, i.e., imaginary time steps.

- The number of samples to discretize each of the up to three dimensions.

- The precision of the used floating point numbers to represent a sample. We used float32 and float64.

## A.1  Hardware Selection and System Details

The measurements for computation speed were done on one server and one desktop computer.

The server is a dual-socket Dell PowerEdge R7525. The two CPUs are AMD Epyc 7543 with 32 Zen 3 cores with eight DDR4-3200 memory channels each. The measurements were limited to one of the CPUs which was configured as a single NUMA node. This sever also contains three NVIDIA A100 80GiB PCIe cards, with one of those used for the measurements. The A100 is a server GPU which was specifically designed for high performance scientific computing and machine learning. It features a peak memory bandwidth of up to 1.94 TB/s and a 1:2 ratio between float32 (19.5 TFLOPS) and float64 (9.7 TFLOPS) performance [58].

The desktop computer consists of an AMD Ryzen 7950X3D CPU and an NVIDIA RTX 4090. The CPU is a high-end desktop CPU with 16 Zen 4 cores and two memory channels of DDR5-5200 RAM. The 16 cores are split into two chiplets with 8 cores each. One chiplet has an additional X3D-Cache which increases its local L3 cache to 96 MiB. The Nvidia RTX 4090 is a high-end consumer GPU. This means it is easier and cheaper to procure and can be used in a normal desktop whereas the A100 is only available for servers. It features also relatively high memory bandwidth (1008 GB/s) but since its focus are graphics workloads, it only has a 1:64 ratio of float64 (1.29 TFLOPS) to float32 (82.6 TFLOPS) performance [47].

Both systems were running Ubuntu 24.04.2 LTS with the NVIDIA driver 570.133.07 and CUDA 12.8. The versions of the used array libraries were NumPy 2.2.6, JAX 0.6.1 and PyTorch 2.7.0. The used version of TorchGPE was the latest public version as of 11 July 2025, `c02428f` on `https://github.com/qo-eth/TorchGPE`.

## A.2   Measurement Methodology

In order to isolate the computational speed in the limit of many time steps, the following procedure was used: First, the simulation is run twice with two time steps in each run as a warm-up to reduce effects due to module imports, compilation caches and other initialization routines which depend on the specific system configuration and software versions.

We only time the inner loop of the simulation without any initialization or clean-up routines while ensuring that after the simulation, the computation results from the GPU have been sent back to the CPU. To reduce run-to-run variations due to interference by other processes, each measurement was done four times and the minimum measured time was taken. This assumes that there is a best case speed and any runtime variation is caused by interruptions which ideally do not happen. To eliminate any startup costs, we use the fact that the simulation time scales linearly in the number of time steps (appendix A.3). For each point in figs. 9 and 11 to 13, we measure for three different numbers of time steps. The base number of steps is adjusted via a calibration run such that the simulation loop takes about ten seconds, the other two time step groups are then two and four times as many time steps which results in a targeted runtime of 20 and 40 seconds, respectively. Each of the points in the diagram figs. 9 and 11 to 13 is then the slope of a linear fit through the minimum runtime for each of the three different numbers of time steps.

## A.3   Scaling in the Number of Time Steps

Theoretically the run time should increase linearly as a function of the number of time steps. As shown in fig. 10 this is indeed the case for the "FFTArray Direct" variant and TorchGPE. Each plot also visualizes the run-to-run variation by plotting all slower runs as grey markers. In most cases, these are not even visible. The points, reduced via taking the minimum, show a clear linear scaling in the number of time steps as shown by the linear fits.

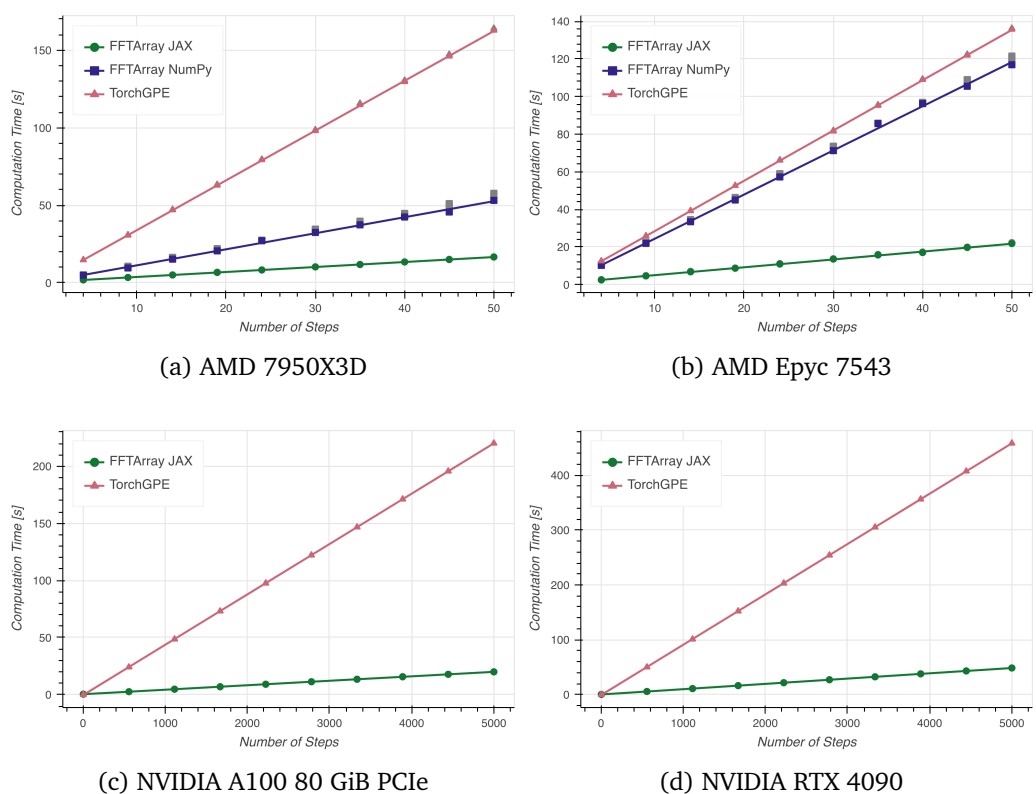

(a) AMD 7950X3D

(b) AMD Epyc 7543

(c) NVIDIA A100 80 GiB PCIe

(d) NVIDIA RTX 4090

Figure 10: The measured run time for an imaginary time evolution of 4096 by 4096 samples in an isotropic harmonic potential as a function of the number of time steps in float64 precision. The used implementations are "FFTArray Direct" for the FFTArray cases and TorchGPE. Grey markers show the durations of the runs which were not selected by the minimum reduction. On most points all measurements are so close together that they overlap with the selected point and are not visible. The lines are linear fits on the minimum run time for each duration. All implementations on all hardware configurations show a clear linear scaling in the number of steps which is also the expected behavior.

## A.4   Scaling in the Number of Samples and Shape of the Domain

The time complexity class of the simulation algorithm as a function of samples is $\mathcal{O}(n_{\text{samples}} \log n_{\text{samples}})$ for the number of samples in each dimension. This comes from the fact that this is the complexity class of the FFT while all other operations are in the class $\mathcal{O}(n_{\text{samples}})$. However, in practice other factors can be more dominant. There is for example a whole hierarchy of memory types with different speed and size. The fastest and smallest are the registers, then there are two to three levels of caches and then the (V)RAM. Additionally, not all memory accesses are equal, since most processors always have to load data at a minimum granularity. Therefore the data layout and the memory access patterns of algorithms can have a huge impact on performance. To get a rough overview of how important these factors are, the "FFTArray Direct" and TorchGPE implementations were tested for different grid sizes and layouts. We always compare the same total number of samples, but these samples are distributed into different shapes:

- 1D: The domain is a single dimension which contains all samples.

- 2D: The domain is two-dimensional with both sides having the same number of samples.

- 3D: The domain is three-dimensional with all three sides having the same number of samples.

- (64, 64, nz): The first and second dimension have 64 samples while the z dimension has a number of samples which is a power of two. This scenario is for example relevant for the case of a Bragg beam splitter. The Bragg beam splitting process requires a higher resolution in the beam direction than the other directions.

- (nx, 64, 64): This is in principle the same shape as the case before. But due to the different ordering of axes, this may change the memory access patters of the algorithm. This case is tested to see if that makes a significant difference.

In order to always test the exact same FFT algorithm, each axis must have a size which is a power of two. Notably, that means that for example the 3D case can only be tested for a total number of samples of $2^{3n}$ with n being a natural number. Additionally we only tested starting from a minimum size of 64 samples per axis and a total minimum number of samples of $2^{20}$ for FFTArray and $2^{18}$ for TorchGPE, since smaller numbers are not that interesting for our applications. TorchGPE only officially supports 2D, so only that shape was tested.

The measurements in this section were done according to appendix A.2 including a linear fit over multiple runs with different numbers of steps.

The results for JAX (fig. 11) and TorchGPE (fig. 13) show a mostly linear scaling for all hardware devices while NumPy (fig. 12) shows a worse than linear scaling, especially in the 1D case. The approximately linear scaling despite the theoretical complexity class of $\mathcal{O}(n_{\text{samples}} \log n_{\text{samples}})$ shows that we are apparently still in a regime of $n_{\text{samples}}$ where other factors dominate the run time. With the exception of the 1D case on CPUs, all layouts show a similar performance, so a good estimation of computational speed can be done simply from the number of samples and time steps. Notably, the 1D case is slower than the other layouts on CPU with both the NumPy and the JAX implementation but on the GPUs all layouts are comparable.

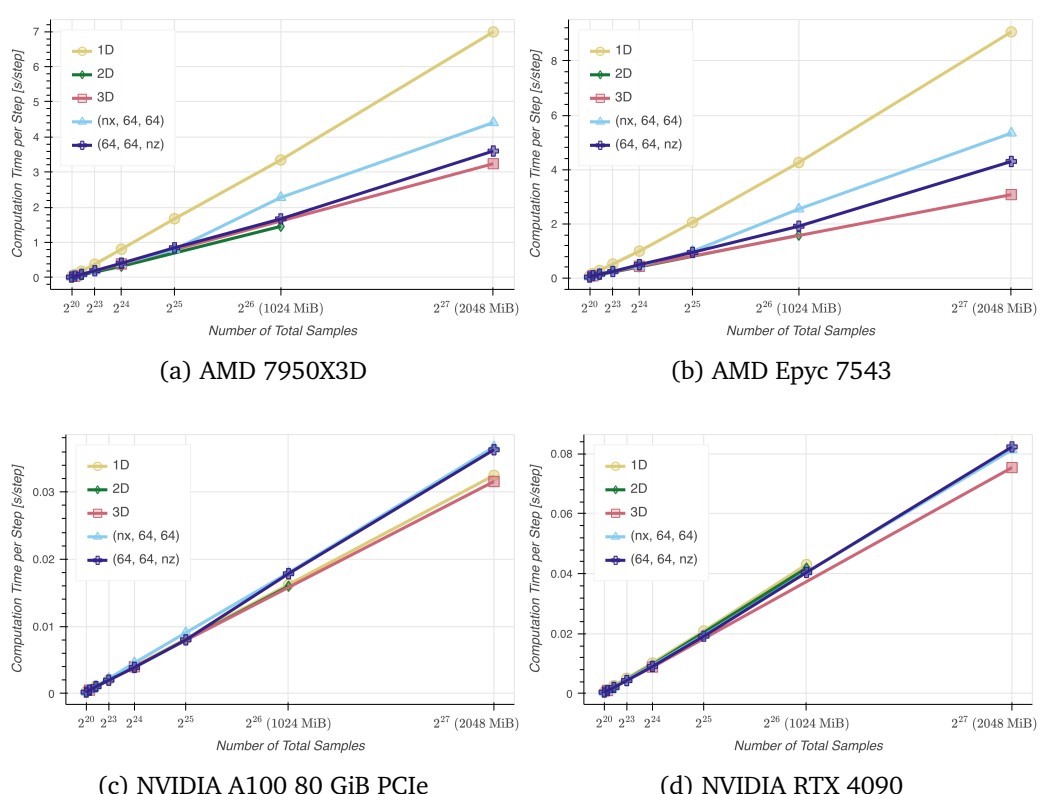

Figure 11: Computation time per step as a function of the number of samples and their layout with the JAX "FFTArray Direct" implementation in float64. Even though the theoretical scaling is $\mathcal{O}(n_{\text{samples}} \log n_{\text{samples}})$, JAX effectively scales linearly in the number of samples. All layouts are reach similar speeds with the exception of the 1D case on CPUs which is significantly slower. On GPUs the 1D case also achieves similar speeds to the other layouts. All points are directly connected for better visual clarity.

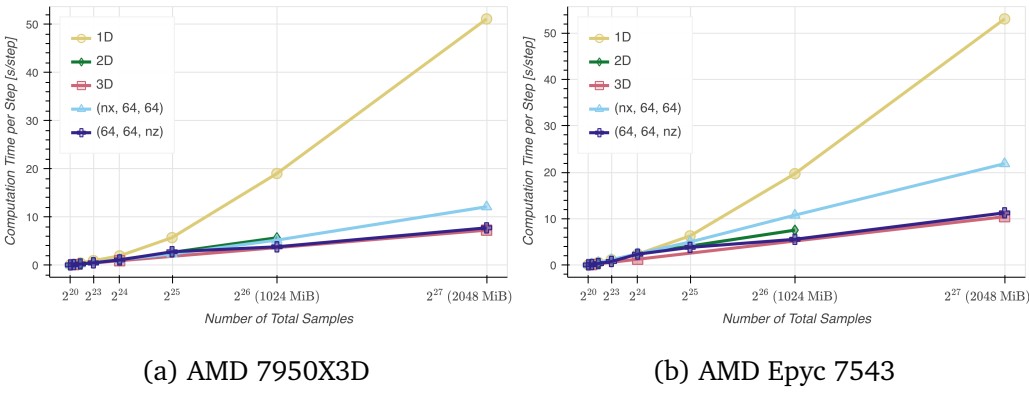

Figure 12: Computation time per step as a function of the number of samples and their layout with the NumPy "FFTArray Direct" implementation in float64. NumPy shows a slower than linear scaling in the number of samples. All layouts are simulated at similar speeds with the exception of the 1D case which is significantly slower. All points are directly connected for better visual clarity.

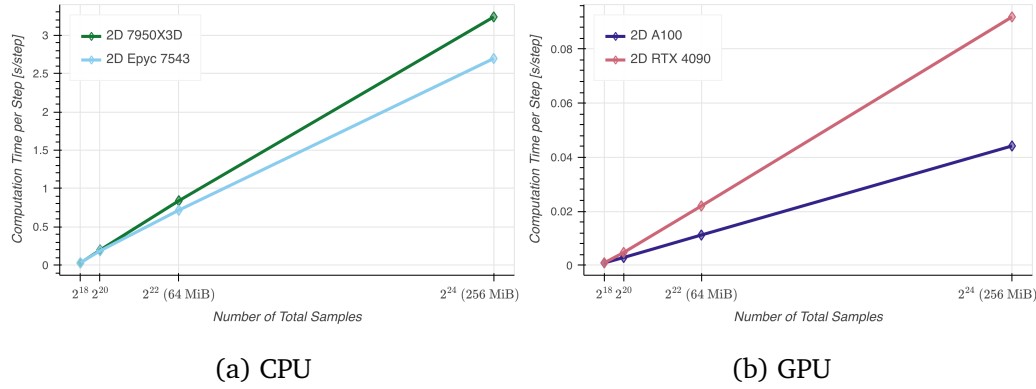

|              |              |
|:------------:|:------------:|
| (a) CPU      | (b) GPU      |

Figure 13: Computation time per step as a function of the number of samples and their layout with the TorchGPE implementation in float64. Even though the theoretical scaling is $\mathcal{O}(n_{\text{samples}} \log n_{\text{samples}})$, TorchGPE effectively scales linearly in the number of samples. Only the 2D configuration was tested since TorchGPE does not support 1D or 3D domains. All points are directly connected for better visual clarity.

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
