# Peer review of "FFTArray: A Python Library for the Implementation of Discretized Multi-Dimensional Fourier Transforms"

_SciPost Physics Codebases_

## Round 2 · Referee Report · Anonymous (Referee 1) · 2025-12-10

Strengths

1- Addresses a common pain area in python programming when utilizing FFTs for customized simulations.
2- Provides a unique and lightweight solution for smooth integration in current workflows.
3- Very well formatted with clear documentation of codebase and examples.
4- Reduces overhead for modifying current FFT implementations (scipy, numpy) and applying the correct transformations for multidimensional transforms.

Weaknesses

1- Introduction on discretizing Fourier transforms needs better discussion on the general existence of the FT for continuous functions and existence constraints with references
2- Section 2.1 implicitly assumes a bandlimited function (wavepacket) which in general is decomposed into an infinite series of waves which will always be undersampled during discretization leading to aliasing. This is not an exact lossless transformation (a sinc function on the other hand is an exactly reconstructable function when undersampled). Discussion of aliasing is not accurate and should be enhanced with specific discussion on choice of sampling rate(spacing) with respect to the continuous function of interest.

Report

Overall, this paper presents an immensely useful python tool capable of simplifying many physics simulations that rely on FFT at some level. The time spent tracking down scale factor and phase factor applications working with other FFT implementations and constructing/reconstruciting position and frequency grids is a severe bottleneck in physics simulation. This tool provides an efficient and general method to focus on the physics and not the computation. While also providing an efficient method of computation with no added overhead on system resources.
Each detail in the presentation is well thought out providing solid background and workable examples throughout. The comparisons to current methodologies using NumPy fft make switching code to FFTArray straightforward.

Requested changes

1- Update description of aliasing and exact transformations when sampling continuous functions in section 2.1.

Recommendation

Publish (meets expectations and criteria for this Journal)

---

## Round 3 · Author Response

Dear Editor, dear Referee,

thank you for your positive assessment. We are grateful for the referee’s feedback, which helped us improve the paper. Following the weaknesses pointed out by the referee we added an additional paragraph about the mathematical foundations of the Fourier Transform and substantially revised section 2.1 to more clearly discuss the sampling theorem, sampling continuous functions and other discretization effects.

We hope these revisions address the concerns satisfactorily. We appreciate the reviewer’s positive evaluation and recommendation.

For the authors,
Stefan J. Seckmeyer

---

## Round 3 · List of Changes

• In section 2 we added an additional paragraph about the mathematical foundations and existence conditions of the Fourier transform with references to Stein and Shakarchi (2003) [17], Hörmander (2003) [18] and Stein and Shakarchi (2005) [19].

  • We substantially revised section 2.1 to clearly separate the sampling theorem from other discretization effects, and we expanded the discussion of sampling continuous functions, the choice of sampling rate and grid size with references to Jerri (1977) [24], Unser(2000) [23], Trefethen (2000) [20], Muga et al (2004) [25], Oppenheim and Schafer (2010) [22], Pharr, Jakob and Humphreys (2023) [21].

  • Adjusted reference ([57] in arXiv:2508.03697v3) to correctly render "10^-17".

---

## Editorial Decision

refereeing_in_preparation